# Prediction of Chronic Atrophic Gastritis and Gastric Neoplasms by Serum Pepsinogen Assay: A Systematic Review and Meta-Analysis of Diagnostic Test Accuracy

**DOI:** 10.3390/jcm8050657

**Published:** 2019-05-10

**Authors:** Chang Seok Bang, Jae Jun Lee, Gwang Ho Baik

**Affiliations:** 1Department of Internal Medicine, Hallym University College of Medicine, Sakju-ro 77, Chuncheon, Gangwon-do 24253, Korea; baikgh@hallym.or.kr; 2Institute of New Frontier Research, Hallym University College of Medicine, Chuncheon 24253, Korea; iloveu59@hallym.or.kr; 3Institute for Liver and Digestive Diseases, Hallym University, Chuncheon 24253, Korea; 4Department of Anesthesiology and Pain Medicine, Hallym University College of Medicine, Chuncheon 24253, Korea

**Keywords:** gastritis, atrophic, pepsinogens, gastric neoplasms

## Abstract

Serum pepsinogen assay (sPGA), which reveals serum pepsinogen (PG) I concentration and the PG I/PG II ratio, is a non-invasive test for predicting chronic atrophic gastritis (CAG) and gastric neoplasms. Although various cut-off values have been suggested, PG I ≤70 ng/mL and a PG I/PG II ratio of ≤3 have been proposed. However, previous meta-analyses reported insufficient systematic reviews and only pooled outcomes, which cannot determine the diagnostic validity of sPGA with a cut-off value of PG I ≤70 ng/mL and/or PG I/PG II ratio ≤3. We searched the core databases (MEDLINE, Cochrane Library, and Embase) from their inception to April 2018. Fourteen and 43 studies were identified and analyzed for the diagnostic performance in CAG and gastric neoplasms, respectively. Values for sensitivity, specificity, diagnostic odds ratio, and area under the curve with a cut-off value of PG I ≤70 ng/mL and PG I/PG II ratio ≤3 to diagnose CAG were 0.59, 0.89, 12, and 0.81, respectively and for diagnosis of gastric cancer (GC) these values were 0.59, 0.73, 4, and 0.7, respectively. Methodological quality and ethnicity of enrolled studies were found to be the reason for the heterogeneity in CAG diagnosis. Considering the high specificity, non-invasiveness, and easily interpretable characteristics, sPGA has potential for screening of CAG or GC.

## 1. Introduction

Gastric cancer (GC) is a global health-related burden and the fourth most common cause of cancer-related deaths worldwide [1]. The sequential cascade of histopathology for development of intestinal-type gastric adenocarcinoma is from normal gastric epithelium to chronic gastritis, chronic atrophic gastritis (CAG), and intestinal metaplasia (IM), followed by dysplasia, and finally GC [2]. Patients with premalignant lesions, such as CAG or dysplasia, have a considerable risk for developing GC, and early detection of these lesions is important for the screening of GC [3,4].

For the population-based screening of GC, the endoscopic mass screening program has shown its efficacy in GC-prevalent countries such as Korea and Japan [5]. The endoscopic screening program reduced GC-related mortality by 47% in a nested case-control study in Korea [6]. However, it is not cost-effective in regions with low incidence of GC, and stepwise or individualized screening according to the risk factors of GC has been recommended [4,5].

In addition to endoscopic diagnosis using visual inspection (with or without image-enhanced endoscopy) or histologic diagnosis using an updated Sydney system for CAG or IM, serum pepsinogen assay (sPGA), which reveals concentration of pepsinogen I (PG I) and ratio of PG I/PG II, has been proposed as a non-invasive test for predicting CAG or GC, reflecting gastric mucosal secretory status [4,7].

Although various cut-off values have been suggested, the combination of PG I ≤70 ng/mL and PG I/PG II ratio ≤3 have been proposed for the prediction of CAG or GC [4,8]. However, previous meta-analyses for diagnostic test accuracy (DTA) presented only pooled outcomes, which cannot determine the diagnostic validity of sPGA with a cut-off value of PG I ≤70 ng/mL and PG I/PG II ratio ≤3 [9,10], although no threshold effect was detected [9]. This can lead to an exaggerated summary of performance because pooled analysis adopted the best performance value in each study, irrespective of cut-off values. Moreover, several articles were omitted during article searching process, and inaccurate calculation of crude values of diagnostic performance, such as true positive (TP), false positive (FP), true negative (TN), and false negative (FN) values, was detected (Table 1).

Another meta-analysis showed higher discriminative efficacy of combining sPGA with *Helicobacter pylori* antibody compared to sPGA or *H. pylori* antibody alone for the prediction of gastric cancer [11]. However, this study presented only comparative efficacy and could not determine the diagnostic validity of each test. Two recently published meta-analyses of DTA showed combined test accuracy of sPGA with *H. pylori* antibody and gastrin-17 for the prediction of CAG (GastroPanel^®^) [12,13]. However, this test is not widely available in Asian countries and also the diagnostic validity of sPGA alone was impossible to determine. Although two earlier meta-analyses presented pooled performance of sPGA with cut-off value of PG I ≤70 ng/mL and PG I/PG II ratio ≤3 [8,14], crude values of diagnostic performance used in these studies are unknown. Moreover, diagnostic values with different cut-off standards were coded as those with PG I ≤70 ng/mL and PG I/PG II ratio ≤3, assuming an intrinsic cut-off effect (Table 1).

Therefore, our study aims to provide evidence of sPGA with cut-off value of PG I ≤70 ng/mL and/or PG I/PG II ratio ≤3 for predicting CAG and gastric neoplasms.

## 2. Materials and Methods

This systematic review and meta-analysis fully adhered to the principles of the Preferred Reporting Items for Systematic reviews and Meta-Analyses for Protocols (PRISMA-P) [15]. The protocol of this study was registered at PROSPERO on December 2018 (registration number, CRD42018116470) before the study was initiated. The approval of institutional review board was exempted as the study collected and synthesized data from published literatures [4].

### 2.1. Literature Searching Strategy

MEDLINE (through PubMed), the Cochrane library, and Embase were searched using common keywords associated with sPGA, CAG, and gastric neoplasms, from the time of inception of these databases to March 2019, by two independent evaluators (C.S.B., and J.J.L). Keywords from Medical Subject Heading and Emtree were selected for searching the electronic databases. Abstracts of all the identified studies were reviewed to exclude irrelevant publications. Full-text reviews were performed to determine whether the inclusion criteria were satisfied throughout all the studies. Bibliographies of relevant articles were rigorously reviewed to identify additional studies. Disagreements between the evaluators were resolved by discussion or consultation with a third evaluator (G.H.B.). The detailed searching strategy is described in Table 2 [4].

### 2.2. Selection Criteria

We included studies that met the following criteria. Patients (1) who have CAG or gastric neoplasms (dysplasia or cancer); (2) intervention: sPGA with cut-off value of PG I ≤70 ng/mL and/or PG I/PG II ratio ≤3; (3) comparison: none; (4) outcome: diagnostic performance indices of sPGA for CAG or gastric neoplasms including sensitivity, specificity, positive predictive value (PPV), negative predictive value (NPV), positive likelihood ratio (PLR), negative likelihood ratio (NLR), accuracy, or diagnostic odds ratio (DOR), which enable an estimation of TP, FP, TN, and FN values; (5) study design: all types; (6) studies of human subjects; and (7) full-text publications written in English. Studies that met all of the inclusion criteria were sought and selected. The exclusion criteria were as follows: (1) narrative review; (2) letter, comment, editorial or reply to questions; (3) study protocol; (4) publication with incomplete data; and (5) systematic review/meta-analysis or consensus report. Studies meeting at least one of the exclusion criteria were excluded from this analysis.

### 2.3. Methodological Quality

The methodological quality of the included publications was assessed using the Quality Assessment of Diagnostic Accuracy Studies-2 (QUADAS-2) tool, which contains four domains, including “patient selection”, “index test”, “reference standard”, and “flow and timing” (flow of patients through the study and timing of the index tests and reference standard) [16]. Each domain is assessed in terms of high-, low-, or unclear risk of bias, and the first three domains are also assessed in terms of high-, low-, or unclear concerns about applicability [16]. Two of the evaluators (C.S.B. and J.J.L.) independently assessed the methodological quality of all the included studies, and any disagreements between the evaluators were resolved by discussion or consultation with a third evaluator (G.H.B.) [4].

### 2.4. Data Extraction and Primary and Modifier-Based Analyses

Two evaluators (C.S.B. and J.J.L.) independently used the same data fill-in form to collect the summary of primary outcomes (TP, FP, FN, and TN) and modifiers in each study. Disagreements between the two evaluators were resolved by discussion or consultation with a third author (G.H.B).

DTA was the primary outcome of this study. We calculated the values for TP (subjects with positive sPGA who have CAG or gastric neoplasms), FP (subjects with positive sPGA who do not have CAG or gastric neoplasms), FN (subjects with negative sPGA who have CAG or gastric neoplasms), and TN (subjects with negative sPGA who do not have CAG or gastric neoplasms) of sPGA for the diagnosis of CAG or gastric neoplasm. To calculate the values, we used 2 × 2 tables whenever possible, from the original articles that contain various diagnostic performance indices (sensitivity, specificity, PPV, NPV, PLR, NLR, accuracy, or DOR etc.). If only a part of data was presented, we calculated the values for TP, FP, FN, and TN using the following formulas: sensitivity = TP/(TP + FN); specificity = TN/(FP + TN); PPV = TP/(TP + FP); NPV = TN/(FN + TN); PLR = sensitivity/(1-specificity); NLR = (1-sensitivity)/specificity; accuracy = (TP + TN)/(TP + FP + FN + TN); DOR = (TP × TN)/(FP × FN); standard error = (ln(upper confidence interval (CI)) – ln(lower CI))/3.92 = √(1/TP + 1/FP + 1/FN + 1/TN).

The following data were also extracted from each study, whenever possible: study design, distribution of age, gender or ethnicity of enrolled population, sample size, published year, measurement method of sPGA, and the proportion of smokers and *H. pylori*-infected individuals.

### 2.5. Statistical Analysis

Stata Statistical Software, version 15.1 (College Station, TX, USA) including relevant packages, such as metandi, midas, and mylabels, was used for this meta-analysis.

Narrative (descriptive) synthesis was planned and quantitative synthesis (bivariate random model [17] and hierarchical summary receiver operating characteristic (HSROC) model [18]) was used if the included studies were sufficiently homogenous. We calculated or extracted TP, FP, FN, and TN values from each study. A Forest plot of pooled sensitivity or specificity using a bivariate model and summary receiver operating characteristic (SROC) curve using a HSROC model were generated. Heterogeneity across the studies was determined by correlation coefficient between logit transformed sensitivity and specificity using bivariate model [17] and asymmetry parameter, β (beta), where β = 0 corresponds to a symmetric receiver operating characteristic (ROC) curve in which the DOR does not vary along the curve by HSROC model. A positive correlation coefficient (greater than 0) and β with significant *p* value (*p* < 0.05) indicate heterogeneity between studies [18,19]. Visual examination of the SROC curve was also performed to find heterogeneity. We also performed meta-regression and subgroup analyses using the modifiers identified during the systematic review to confirm robustness of the main result and to identify the reason of heterogeneity. Publication bias was evaluated using Deeks’ funnel plot asymmetry test.

## 3. Results

### 3.1. Identification of Relevant Studies

Figure 1 presents a flow diagram showing the process to identify the relevant studies. For CAG diagnosis, a total of 855 articles were identified by searching four electronic databases and additional hand-searching. Among those articles, 174 were duplicate studies, and 552 additional studies were excluded during the initial screening by reviewing titles and abstracts. Full texts of the remaining 129 studies were then thoroughly reviewed. Among these studies, 115 articles were excluded from the final analysis due to the following reasons: narrative review article (*n =* 13), letter, comment, editorial or reply to questions (*n =* 1), study protocol (*n =* 1), incomplete data (*n =* 92), and systematic review/meta-analysis or consensus report (*n =* 8). The remaining 14 studies [20,21,22,23,24,25,26,27,28,29,30,31,32,33] were included in the quantitative synthesis. Eight studies [20,23,24,27,29,30,31,32] adopted the cut-off standard of PG I ≤70 ng/mL and PG I/PG II ratio ≤3, eight studies [21,22,25,26,27,28,30,33] adopted the cut-off standard of PG I/PG II ratio ≤3, and only two studies [27,33] adopted the cut-off standard of PG I ≤70 ng/mL.

For gastric neoplasm diagnosis, a total of 1408 articles were identified by searching four electronic databases. Among those articles, 538 were duplicate studies, and 685 additional studies were excluded during the initial screening by reviewing titles and abstracts. The full texts of the remaining 185 studies were then thoroughly reviewed. Among these studies, 142 articles were excluded from the final analysis, due to the following reasons: narrative review article (*n =* 30), letter, comment, editorial or reply to questions (*n =* 6), study protocol (*n =* 6), incomplete data (*n =* 88), and systematic review/meta-analysis or consensus report (*n =* 12). The remaining 43 studies [21,26,27,34,35,36,37,38,39,40,41,42,43,44,45,46,47,48,49,50,51,52,53,54,55,56,57,58,59,60,61,62,63,64,65,66,67,68,69,70,71,72,73], including 38 studies [26,34,35,36,37,38,39,40,41,42,43,44,45,46,47,48,49,50,51,52,53,54,56,57,58,59,60,62,63,64,65,66,68,69,70,71,72,73] evaluating the performance of sPGA for the diagnosis of GC, four studies [21,27,45,73] for the diagnosis of gastric dysplasia, and four studies [55,61,67,71] for the diagnosis of gastric neoplasm, were incorporated in the quantitative synthesis.

Among the 38 studies for the diagnosis of GC, 28 studies [35,36,37,38,40,41,42,43,46,48,49,50,51,53,54,56,57,58,59,60,62,63,65,66,70,71,72,73] adopted the cut-off standard of PG I ≤70 ng/mL and PG I/PG II ratio ≤3, 11 studies [26,38,39,44,45,47,52,57,65,68,69] adopted the cut-off standard of PG I/PG II ratio ≤3, and only six studies [34,45,57,64,65,69] adopted the cut-off standard of PG I ≤70 ng/mL.

Among the 28 studies that adopted the cut-off standard of PG I ≤70 ng/mL and PG I/PG II ratio ≤3, two studies [46,51] evaluated diagnostic performance for sPGA based on same population with slightly different inclusion criteria. Therefore, to avoid dependence issue from single population-based multiple outcomes, the study with larger population [46] was included in the meta-analysis as a representative outcome. Finally, 27 studies were included for the diagnosis of GC with the cut-off standard of PG I ≤70 ng/mL and PG I/PG II ratio ≤3.

### 3.2. Characteristics of the Included Studies

From the 14 studies [20,21,22,23,24,25,26,27,28,29,30,31,32,33] for the diagnosis of CAG, we identified a total of 5541 patients (2220 patients with CAG vs. 3321 patients without CAG). Among them, 11 [20,22,23,24,25,26,27,28,30,31,32] were case-control studies, whereas two [21,33] were cross-sectional studies and only one [29] was cohort study. Seven studies [20,24,28,29,30,31,33] were conducted in Asia, whereas the remaining studies were conducted in Europe [21,22,26,27,32] and South America [23]. In 2009, Leja M et al. [25] reported enrollment of the population in Latvia, Lithuania, and Taiwan as an international study setting. The mean age of the enrolled population ranged from 43.6 to 66.3 years. Male predominance was detected in four studies [24,26,28,31], whereas the remaining studies showed female predominance. For detection method of sPGA, most studies used enzyme-linked immunosorbent assay (ELISA), whereas three studies [28,29,32] used latex-enhanced turbidimetric immunoassay (L-TIA) and two studies [20,26] deployed radioimmunoassay (RIA) (Table 3).

From the 43 studies [21,26,27,34,35,36,37,38,39,40,41,42,43,44,45,46,47,48,49,50,51,52,53,54,55,56,57,58,59,60,61,62,63,64,65,66,67,68,69,70,71,72,73] for the diagnosis of gastric neoplasm, we identified a total of 114,448 patients (4689 patients with GC, 430 patients with neoplasm, 130 patients with dysplasia vs. 109,199 patients without gastric neoplasm). Among them, 20 [26,27,34,36,37,40,43,44,45,54,55,56,57,58,61,62,64,67,68,72] were case–control studies, whereas 18 [35,39,41,42,46,47,48,49,50,51,53,59,60,63,66,69,70,71] were cohort studies, and only five [21,38,52,65,73] were cross-sectional studies. Six studies [21,26,27,59,70,73] were conducted in Europe, whereas the remaining 37 studies were conducted in Asia. The mean age of the enrolled population ranged from 33.4 to 68.2 years. Most of the studies showed male predominance except 11 studies [21,27,38,42,53,59,60,66,67,70,73] showing female predominance. For detection of sPGA, most studies used RIA, whereas nine studies [21,27,50,54,59,64,65,69,70] used ELISA, eight studies [45,52,55,61,67,71,72,73] used L-TIA, four studies [49,56,57,62] used chemiluminescent immunoassay (CLIA), two studies [43,68] used enzyme immunoassay (EIA), and one study [48] used either RIA or L-TIA (Table 4).

### 3.3. Methodological Quality of the Include Studies

Methodological qualities of the included studies were similar for the diagnosis of CAG except for five studies. Most of the studies used histological diagnosis as a reference standard of CAG diagnosis; however, three studies [20,28,31] deployed endoscopic diagnosis (visual inspection) as a reference standard of CAG diagnosis. One study [21] included only high-risk patients, such as patients with severe CAG, IM, and dysplasia, excluding the healthy population. Another study [33] also included high-risk patients as a population for reference standard. These five studies for the diagnosis of CAG were rated as “high-risk” in at least one of the seven domains (Figure 2).

Methodological qualities of the included studies were similar for the diagnosis of gastric neoplasm except for 13 studies. Ideally, all the patients should be tested with the same reference standard method (endoscopy). However, seven studies [35,39,46,47,49,51,63] performed endoscopy to diagnose gastric neoplasm only for patients with positive sPGA or positive double-contrast barium X-ray introducing partial verification bias. One study [48] conducted endoscopy every 2 years for patients with positive sPGA and every 5 years for patients with negative sPGA, adopting different standards of reference test (differential verification bias).

Five studies [39,46,47,51,63] included only male patients, one study [44] included only patients with *H. pylori* infection, two studies [57,72] included only patients with early GC, and one study [58] included only patients with diffuse-type GC.

Two studies [46,51] evaluated diagnostic performance of sPGA based on a same population with slightly different inclusion criteria and another two studies [46,47] also evaluated diagnostic performance based on a same population using different cut-off values. Therefore, these studies were ranked as “high-risk” for the applicability concerns.

Since most of the studies were case-control studies, they were not ranked as “high-risk”. A total of 13 abovementioned studies for the diagnosis of gastric neoplasm were rated as “high-risk” in at least one of the seven domains (Figure 3).

### 3.4. DTA of sPGA in CAG

Values for sensitivity, specificity, PLR, NLR, DOR, and area under the curve (AUC) with 95% CI for the cut-off value of PG I ≤70 ng/mL and PG I/PG II ratio ≤3 for CAG diagnosis were 0.59 (95% CI: 0.38–0.78), 0.89 (0.70–0.97), 5.5 (2.3–13.0), 0.46 (0.30–0.69), 12 (6–25), and 0.81 (0.77–0.84), respectively (Table 5, Figure 4A). The SROC curve with 95% confidence region and prediction region is illustrated in Figure 5A. To investigate the clinical utility of sPGA, Fagan’s nomogram was generated. Assuming 20% prevalence of CAG (prior probability), Fagan’s nomogram shows that the posterior probability of CAG is 58% if patients are diagnosed as positive, and the posterior probability of CAG is 10% if patients are diagnosed as negative according to the sPGA with the cut-off value of PG I ≤70 ng/mL and PG I/PG II ratio ≤3 (Figure 6A).

Values of sensitivity, specificity, PLR, NLR, DOR, and AUC with 95% CI for the cut-off value of PG I/PG II ratio ≤3 for CAG diagnosis were 0.50 (0.28–0.72), 0.94 (0.82–0.98), 7.8 (3.3–18.1), 0.53 (0.34–0.82), 15 (6–37), and 0.85 (0.81–0.88), respectively (Table 5, Figure 4B). The SROC curve with 95% confidence region and prediction region is illustrated in Figure 5B. Fagan’s nomogram shows that the posterior probability of CAG is 66% if patients are diagnosed as positive, and the posterior probability of CAG is 12% if patients are diagnosed as negative according to the sPGA with the cut-off value of PG I/PG II ratio ≤3 (Figure 6B).

### 3.5. DTA of sPGA in GC

Since the minimum number of studies required for the quantitative analysis is four, DTA summary of sPGA in dysplasia or neoplasm was not calculated with a specific cut-off standard (only two or three studies were included with a specific cut-off value) (Figure 1).

Sensitivity, specificity, PLR, NLR, DOR and AUC with 95% CI for the cut-off value of PG I ≤70 ng/mL and PG I/PG II ratio ≤3 for GC diagnosis were 0.59 (0.50–0.67), 0.73 (0.64–0.81), 2.2 (1.7–2.9), 0.56 (0.46–0.68), 4 (3–6), and 0.70 (0.66–0.74), respectively (Table 6, Figure 7A). The SROC curve with 95% confidence region and prediction region is illustrated in Figure 8A. Assuming 20% prevalence of GC (prior probability), Fagan’s nomogram shows that the posterior probability of GC is 36% if patients are diagnosed as positive, and the posterior probability of GC is 13% if patients are diagnosed as negative according to the sPGA with the cut-off value of PG I ≤70 ng/mL and PG I/PG II ratio ≤3 (Figure 9A).

Values for sensitivity, specificity, PLR, NLR, DOR, and AUC with 95% CI for the cut-off value of PG I ≤70 ng/mL for GC diagnosis were 0.62 (0.38–0.82), 0.57 (0.32–0.79), 1.4 (0.9–2.3), 0.67 (0.40–1.11), 2 (1–5), and 0.63 (0.58–0.67), respectively (Table 6, Figure 7B). The SROC curve with 95% confidence region and prediction region is illustrated in Figure 8B. Fagan’s nomogram shows that the posterior probability of GC is 26% if patients are diagnosed as positive, and the posterior probability of GC is 14% if patients are diagnosed as negative according to the sPGA with the cut-off value of PG I ≤70 ng/mL (Figure 9B).

Values for sensitivity, specificity, PLR, NLR, DOR, and AUC with 95% CI for the cut-off value of PG I/PG II ratio ≤3 for GC diagnosis were 0.56 (0.35–0.75), 0.78 (0.62–0.88), 2.5 (1.7–3.7), 0.56 (0.39–0.81), 4 (3–8), and 0.74 (0.70–0.78), respectively (Table 6, Figure 7C). The SROC curve with 95% confidence region and prediction region is illustrated in Figure 8C. Fagan’s normogram shows that the posterior probability of GC is 39% if patients are diagnosed as positive, and the posterior probability of GC is 12% if patients are diagnosed as negative according to the sPGA with the cut-off value of PG I/PG II ratio ≤3 (Figure 9C).

### 3.6. Exploring Heterogeneity with Meta-Regression and Subgroup Analysis of sPGA in CAG

For the diagnosis of CAG with the cut-off value of PG I ≤70 ng/mL and PG I/PG II ratio ≤3, the SROC curve was symmetric (Figure 5A). We observed a negative correlation coefficient between logit transformed sensitivity and specificity (−0.92) and asymmetry parameter, β, with non-significant *p* value (*p =* 0.14) indicating no heterogeneity among studies. However, 95% prediction region in the SROC curve was wide, and age (*p =* 0.01) and methodological quality of the included studies (*p =* 0.01) were found to be the source of heterogeneity in meta-regression. Subgroup analyses according to the modifiers of heterogeneity showed lower AUCs in studies with a younger population (<60 years) and high methodological quality (Table 5).

For the diagnosis of CAG with the cut-off value of PG I/PG II ratio ≤3, the SROC curve was symmetric (Figure 5B). We observed a negative correlation coefficient between logit transformed sensitivity and specificity (−0.72) and asymmetry parameter, β, with non-significant *p* value (*p =* 0.70), indicating no heterogeneity among studies. However, the 95% prediction region in the SROC curve was wide, and ethnicity (*p =* 0.02), age (*p =* 0.03), methodological quality of included studies (*p =* 0.01), and total number of patients (*p =* 0.05) were found to be the source of heterogeneity in meta-regression. Subgroup analyses according to the modifiers of heterogeneity showed lower AUCs in studies with a younger population (<60 years), an Asian population, low methodological quality, and higher number of included patients (≥1000) (Table 5).

### 3.7. Exploring Heterogeneity with Meta-Regression and Subgroup Analysis of sPGA in GC

For the diagnosis of GC with the cut-off value of PG I ≤70 ng/mL and PG I/PG II ratio ≤3, SROC curve was symmetric (Figure 8A). We observed a negative correlation coefficient between logit transformed sensitivity and specificity (−0.38) and asymmetry parameter, β, with non-significant *p* value (*p =* 0.26), indicating no heterogeneity among studies. However, 95% prediction region in SROC curve was wide and ethnicity (*p =* 0.02), published year (*p =* 0.01), and total number of patients (*p =* 0.01) were found to be the source of heterogeneity in meta-regression. Subgroup analyses according to the modifiers of heterogeneity showed lower AUCs in studies with Western population, more recent publications (2010–2018 vs. 1995–2009) and lower number of included patients (<1000) (Table 6).

For the diagnosis of GC with the cut-off value of PG I ≤70 ng/mL, the SROC curve was symmetric (Figure 8B). We observed a negative correlation coefficient between logit transformed sensitivity and specificity (−0.61) and asymmetry parameter, β, with non-significant *p* value (*p =* 0.92), indicating no heterogeneity among studies. However, 95% prediction region in the SROC curve was wide and methodological quality of included studies (*p =* 0.05), detection method of sPGA (*p* <0.01), and total number of patients (*p* <0.01) were found to be the source of heterogeneity in meta-regression. Subgroup analyses according to the modifiers of heterogeneity was only possible for methodological quality, because the number of subgroups classified according to the other modifiers was lower than four. Subgroup analysis showed lower AUCs in studies with high methodological quality (Table 6).

For the diagnosis of GC with the cut-off value of PG I/PG II ratio ≤3, the SROC curve was symmetric (Figure 8C). We observed a negative correlation coefficient between logit transformed sensitivity and specificity (−0.83) and asymmetry parameter, β, with non-significant *p* value (*p =* 0.57), indicating no heterogeneity among studies. Only ethnicity (*p* <0.01) was found to be the source of heterogeneity in meta-regression. Subgroup analyses according to the modifier of heterogeneity showed lower AUCs in studies with Asian populations (Table 6).

### 3.8. Publication Bias

Publication bias was not evaluated for diagnosis of CAG, as fewer than 10 studies on this subject were included with any cut-off values.

For the diagnosis of GC, 27 studies were included with cut-off value of PG I ≤70 ng/mL and PG I/PG II ratio ≤3. Deeks’ funnel plot asymmetry test showed no evidence of publication bias (*p =* 0.71) (Figure 10A). Publication bias was not evaluated for cut-off of PG I ≤70 ng/mL, as only six studies were included with this cut-off value. Eleven studies were included with cut-off of PG I/PG II ratio ≤3. Although Deeks’ funnel plot asymmetry test for 11 studies with a cut-off value of PG I/PG II ratio ≤3 showed a *p* value of 0.02, indicating publication bias, the plot was symmetrical with respect to the regression line (Figure 10B).

## 4. Discussion

There are two main types of pepsinogen (PG), namely PG I and PG II, which are proenzymes of pepsin, an endoproteinase present in the gastric juice [21]. PG I is secreted mainly by chief cells in the fundic glands of the stomach fundus and body, whereas PG II is secreted by all the gastric glands and the proximal duodenal mucosa (Brunner’s glands) [5,21,74,75]. The secretion ability of gastric mucosa is usually intact in the case of no infection or acute *H. pylori* infection [75]. However, when chronic *H. pylori* infection with CAG extends from antrum to corpus of stomach, chief cells are replaced by pyloric glands [7]. Therefore, concentration of serum PG I decreases due to the damaged secretion ability of gastric mucosa, whereas the concentration of PG II remains relatively intact, leading to a low PG I/PG II ratio and this value reflects the severity of CAG [4,7,75].

Although various cut-off values have been suggested, PG I ≤70 ng/mL and PG I/PG II ratio ≤3 have been proposed for the prediction of CAG or GC [4,8]. However, previous meta-analyses presented only pooled outcomes, which cannot determine the diagnostic validity of sPGA with cut-off value of PG I ≤70 ng/mL and PG I/PG II ratio ≤3 [9,10], although no threshold effect was detected [9]. Moreover, the meta-analysis determined publication bias with Begg’s test, which is inappropriate for DTA because of type I error inflation [9]. Serum concentration of gastrin, which is produced and secreted primarily by the G cells in antrum, is increased when the corpus mucosa is predominantly involved, and decreased with antral predominant gastric atrophy [5,75]. Combined efficacy of sPGA with *H. pylori* antibody [11] and/or gastrin-17 [12,13] has been indicated for the prediction of gastric cancer [11] and CAG [12,13], and it is mainly used in Europe (as panel test). However, sPGA is preferable to serum gastrin measurement because sPGA reflects gastric mucosal status better [75]. Moreover, previous meta-analyses could not determine the diagnostic validity of sPGA alone [11,12,13]. Although previous meta-analyses, published in 2004 and 2006, reported diagnostic validity of sPGA with cut-off value of PG I ≤70 ng/mL and PG I/PG II ratio ≤3, these studies cannot reflect recently published data and had several methodological pitfalls [8,14] (Table 1).

The results of our study confirm that the performance of sPGA is better for the diagnosis of CAG than GC, and sPGA has potential for CAG or GC screening (triage test) considering its high specificity (Table 5 and Table 6). Another finding of this study is the diagnostic validity of sPGA with cut-off value of PG I/PG II ratio ≤3. Although direct comparison of DOR does not have significant implications, the DTA of sPGA with cut-off value of PG I/PG II ratio ≤3 was similar to that with cut-off value of PG I ≤70 ng/mL and PG I/PG II ratio ≤3 (Table 5 and Table 6). A recent study also indicated that the PG I/PG II ratio is one of the stomach-specific circulating biomarkers for GC risk assessment [69]. It is also known that sPGA is a cost-effective diagnostic test and useful to reduce the intestinal-type GC, especially for high-risk populations [76,77]. Considering the non-invasiveness and easily interpretable characteristics, the results of this study indicates the utility of sPGA as a population-based screening tool for CAG or GC.

Compared to the previous meta-analyses that combined the diagnostic values with various cut-off standards in a single outcome, the results of this study showed slightly lower diagnostic values (AUC for the diagnosis of CAG: 0.81 vs. 0.85/AUC for the diagnosis of GC: 0.70 vs. 0.76/DOR for the diagnosis of GC: 4 vs. 5.41), indicating overestimation of diagnostic validity in previous studies [9,10].

In terms of the reasons of heterogeneity, subgroup analyses showed decreased *I^2^* values in high-quality studies with cut-off value of PG I ≤70 ng/mL and PG I/PG II ratio ≤3 for the diagnosis of CAG compared to those of main analysis (*I^2^* of sensitivity: 96.4% to 61.5%, *I^2^* of specificity: 96.1% to 88.6%) (Figure 4 and Table 5). In the subgroup analyses with a cut-off value of PG I/PG II ratio ≤3 for the diagnosis of CAG, high-quality studies (*I^2^* of sensitivity: 98.3% to 88.5%, *I^2^* of specificity: 98.2% to 74.2%) and a Western population (*I^2^* of sensitivity: 98.3% to 85.1%, *I^2^* of specificity: 98.2% to 80%) also showed decreased *I^2^* values compared to those of main analysis, indicating needs for high-quality studies with a Western population to enhance the evidence level in this topic. Although studies with Western population showed slightly higher AUC (0.88 vs. 0.85) than pooled AUC, the value is closer to that of high-quality studies subgroup (0.92), indicating it is not an overestimation, rather we need more Western population data to enhance the level of evidence. In Table 6, recently published subgroup showed much lower AUC (0.61 vs. 0.76) than that of old publications; however, the AUC of recently published subgroup was closer to that of high-quality subgroup (0.68; data not shown because it was not a source of heterogeneity in meta-regression), indicating overestimation of older publications. There was a change in diagnostic values according to the modifiers in the subgroup analyses for the diagnosis of GC; however, such decrease of *I^2^* values in the subgroup analyses was not detected (Table 6) (data about *I^2^* not shown in the results section).

The distribution of CAG or IM (known as pre-malignant or high-risk lesions of GC) in entire population affects the determination of optimal cut-off value of sPGA (spectrum bias). In our meta-analysis, a study by Dinis-Ribeiro et al. [21] included high-risk patients of GC, such as those with AG, IM, or dysplasia, excluding the healthy population, and showed higher sensitivity compared to that of pooled analysis with cut-off of PG I/PG II ratio ≤3 (0.66 vs. 0.50) (Table 5). A previous study by Valli De Re et al. [78] also included high-risk patients, such as first-degree relatives of patients with GC or CAG, and showed high sensitivity and specificity of 0.96 and 0.93 for the prediction of Operative Link on Gastric Intestinal Metaplasia Assessment (OLGIM) stage ≥2 with cut-off of PG I ≤47.9 ng/mL. The proposed cut-off of PG I was lower than 70 ng/mL because they included a high-risk population. However, they proposed algorithm approach of using gastrin-17 first, because they included high-risk patients and gastrin-17 showed highest discrimination capacity of CAG among proposed biomarkers. For the next-step, they recommended using PG I ≤47.9 ng/mL for the prediction of OLGIM stage ≥2. PG I generally shows a low level in CAG; however, if an optimal cut-off should be determined in a high-risk population, lower cut-off value might be required. A combination with a marker, such as gastrin-17, which shows high discriminative performance of CAG, could be considered.

The present study rigorously investigated the diagnostic validity of sPGA with well-known cut-off value of PG I ≤70 ng/mL and/or PG I/PG II ratio ≤3 for the diagnosis of CAG or GC, excluding threshold effect. However, the study has several limitations. Firstly, a relatively small number of studies were enrolled with cut-off value of PG I ≤70 ng/mL or PG I/PG II ratio ≤3 compared to the combination of both values. Secondly, potential publication bias was suspected in the diagnosis of GC with cut-off value of PG I/PG II ratio ≤3 (Deeks’ funnel plot asymmetry test showed *p* value of 0.02, although the plot showed symmetrical shape), probably due to relatively small number of enrolled studies (*n =* 11) (Figure 10B). Thirdly, substantial heterogeneity among studies were suspected, although rigorous subgroup analyses were performed and interpreted. Fourthly, this meta-analysis included many case–control studies, which easily overestimate the diagnostic validity of the index test. Fifthly, the diagnostic validity of sPGA is known to be associated with the smoking, *H. pylori* infection status, or the proportion of diffuse-type GC of the enrolled population [79]. However, this information was presented only in small portion of enrolled studies, limiting further analysis.

In conclusion, sPGA has the potential for use as a CAG or GC screening (triage test). Considering the heterogeneity among studies found in this analysis, high-quality studies based on Western populations could enhance the evidence level in this topic. Most importantly, considering that the usefulness of sPGA may be different between countries, this biomarker should be validated before practically using it for the screening of CAG or GC, because the enrolled studies were conducted in only a few countries.

## Figures and Tables

**Figure 1 jcm-08-00657-f001:**
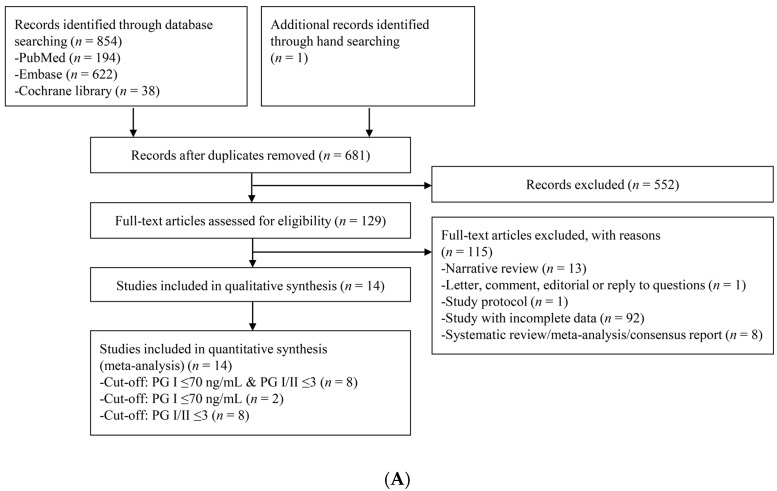
Flow diagram of the identification of relevant studies. (**A**) For the diagnosis of CAG, (**B**) For the diagnosis of gastric neoplasm. PG, pepsinogen; CAG, chronic atrophic gastritis.

**Figure 2 jcm-08-00657-f002:**
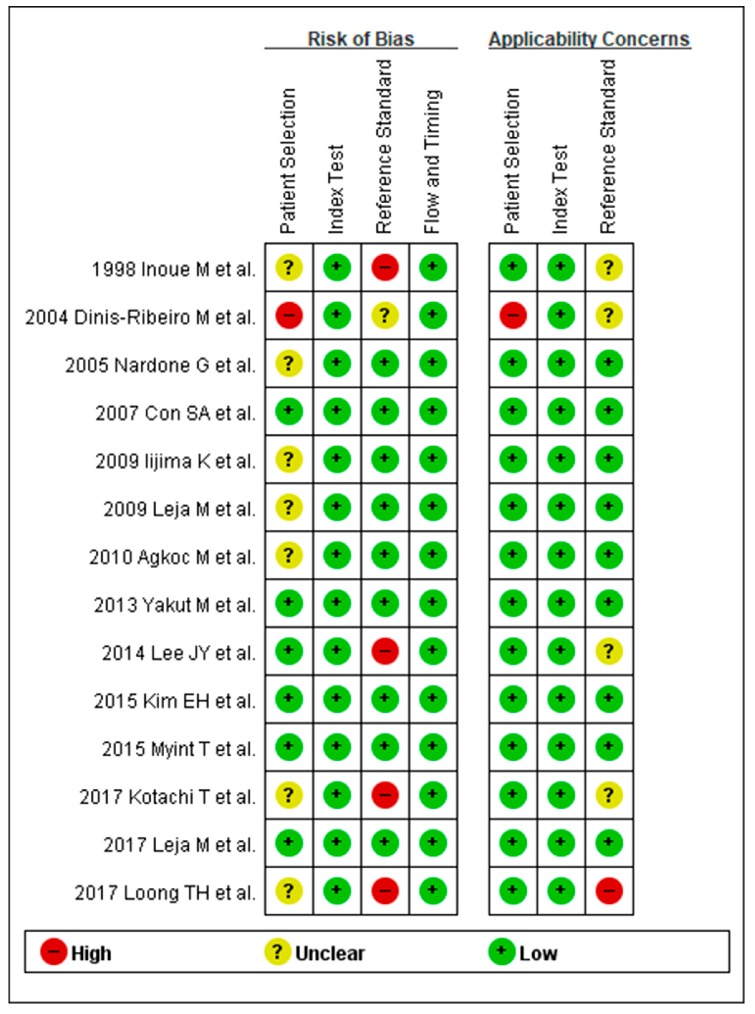
QUADAS-2 for the assessment of the methodological qualities of all the enrolled studies for the diagnosis of CAG. (+) denotes low risk of bias, (?) denotes unclear risk of bias, (–) denotes high risk of bias. QUADAS-2, Quality Assessment of Diagnostic Accuracy Studies-2; CAG, chronic atrophic gastritis.

**Figure 3 jcm-08-00657-f003:**
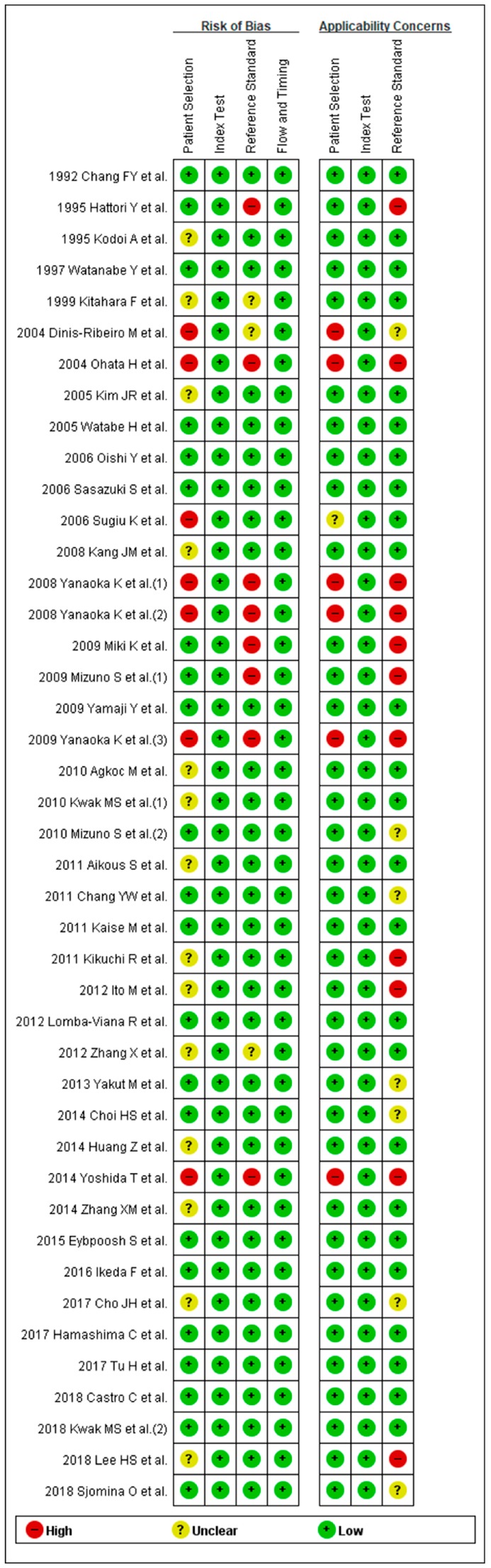
QUADAS-2 for the assessment of the methodological qualities of all the enrolled studies for the diagnosis of gastric neoplasm. (+) denotes low risk of bias, (?) denotes unclear risk of bias, (–) denotes high risk of bias. QUADAS-2, Quality Assessment of Diagnostic Accuracy Studies-2.

**Figure 4 jcm-08-00657-f004:**
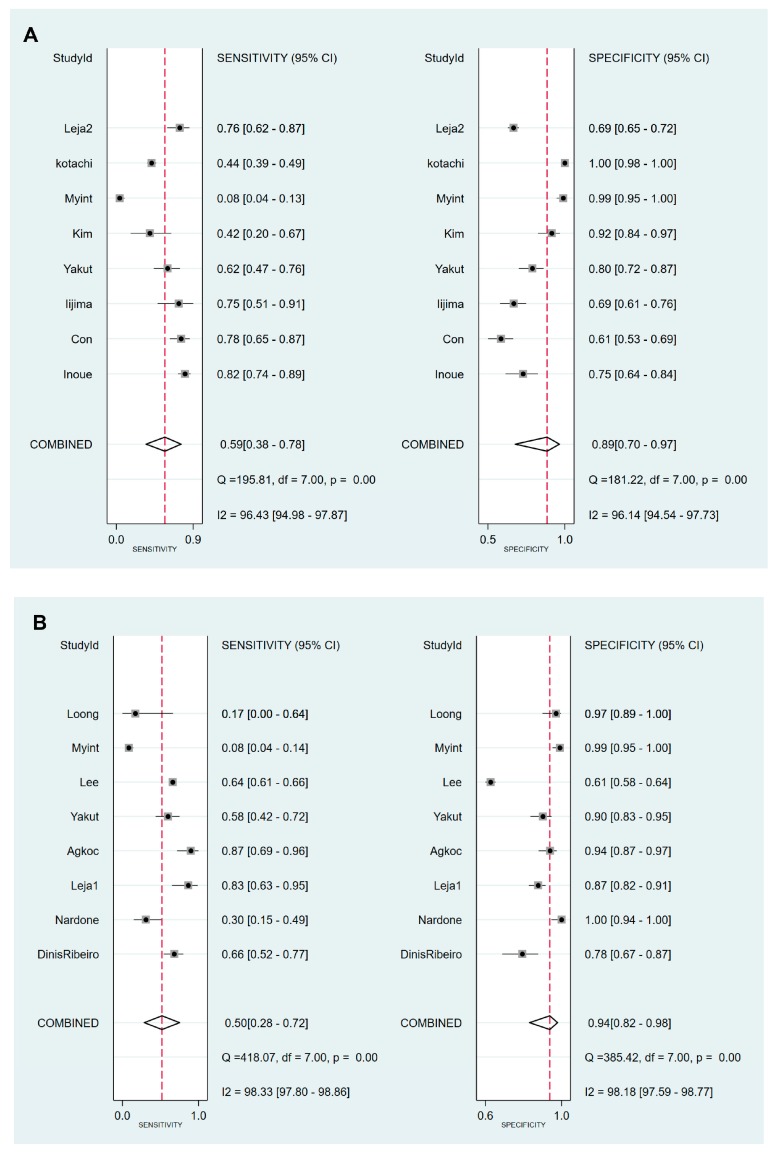
Forest plots of sensitivity and specificity for the diagnosis of CAG. (**A**) cut-off value with PG I ≤70 ng/mL and PG I/PG II ratio ≤3, (**B**) cut-off value with PG I/PG II ratio ≤3. CAG, chronic atrophic gastritis; PG, pepsinogen.

**Figure 5 jcm-08-00657-f005:**
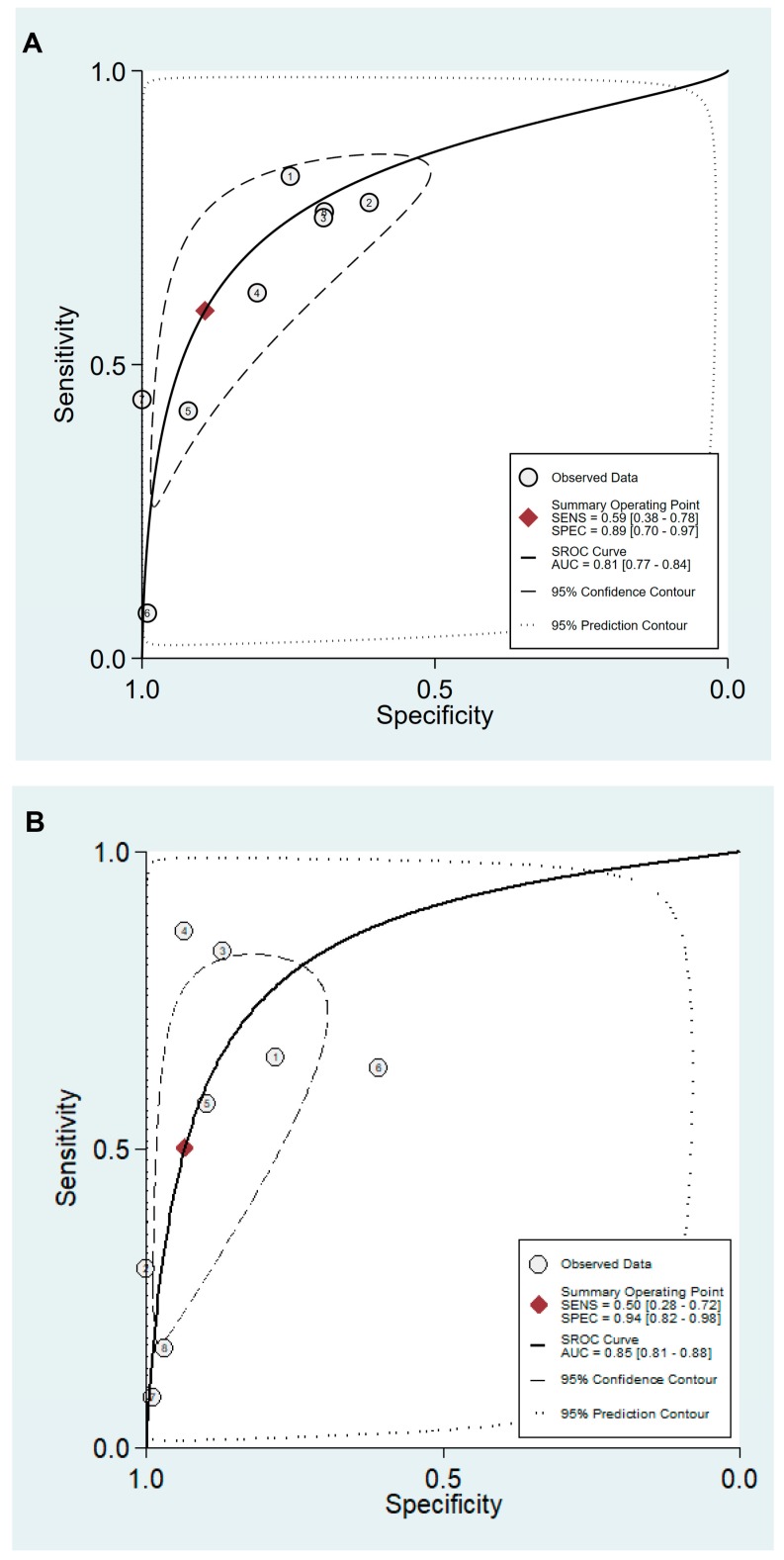
SROC curve with 95% confidence region and prediction region for the diagnosis of CAG. (**A**) cut-off value with PG I ≤70 ng/mL and PG I/PG II ratio ≤3, (**B**) cut-off value with PG I/PG II ratio ≤3. SROC, summary receiver operating characteristic; CAG, chronic atrophic gastritis; PG, pepsinogen; SENS, sensitivity; SPEC, specificity; AUC, area under the curve.

**Figure 6 jcm-08-00657-f006:**
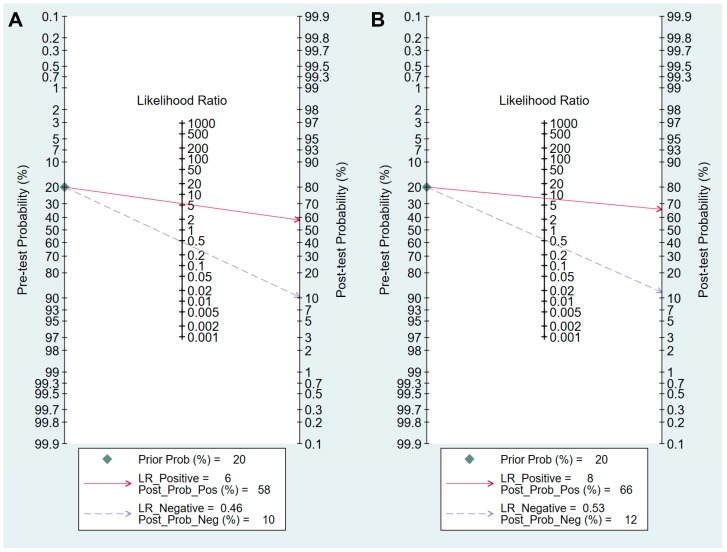
Fagan’s normogram for the diagnosis of CAG. (**A**) cut-off value with PG I ≤70 ng/mL and PG I/PG II ratio ≤3, (**B**) cut-off value with PG I/PG II ratio ≤3. CAG, chronic atrophic gastritis; PG, pepsinogen; LR, likelihood raio.

**Figure 7 jcm-08-00657-f007:**
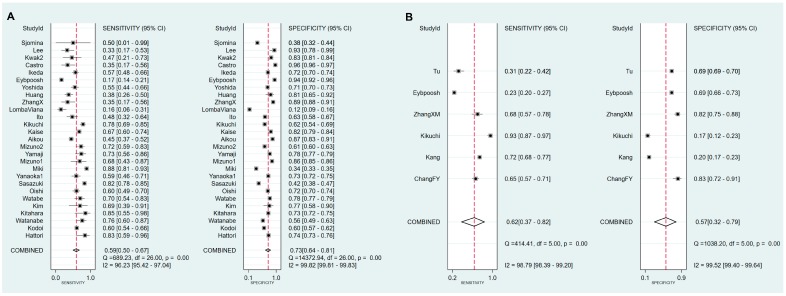
Forest plots of sensitivity and specificity for the diagnosis of GC. (**A**) cut-off value with PG I ≤70 ng/mL and PG I/PG II ratio ≤3, (**B**) cut-off value with PG I ≤70 ng/mL, (**C**) cut-off value with PG I/PG II ratio ≤3. GC, gastric cancer; PG, pepsinogen.

**Figure 8 jcm-08-00657-f008:**
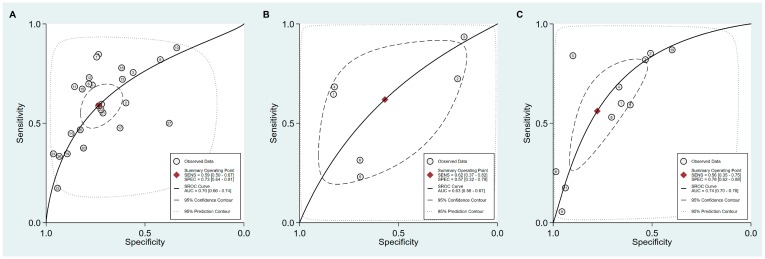
SROC curve with 95% confidence region and prediction region for the diagnosis of GC. (**A**) cut-off value with PG I ≤70 ng/mL and PG I/PG II ratio ≤3, (**B**) cut-off value with PG I ≤70 ng/mL, (**C**) cut-off value with PG I/PG II ratio ≤3. SROC, summary receiver operating characteristic; GC, gastric cancer; PG, pepsinogen.

**Figure 9 jcm-08-00657-f009:**
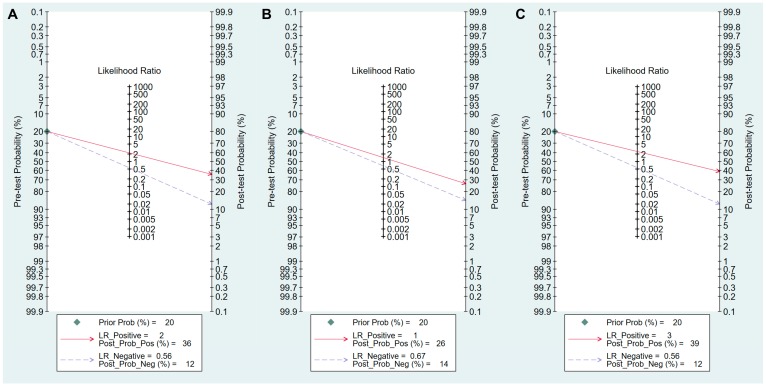
Fagan’s normogram for the diagnosis of GC. (**A**) cut-off value with PG I ≤70 ng/mL and PG I/PG II ratio ≤3, (**B**) cut-off value with PG I ≤70 ng/mL, (**C**) cut-off value with PG I/PG II ratio ≤3. GC, gastric cancer; PG, pepsinogen; LR, likelihood raio.

**Figure 10 jcm-08-00657-f010:**
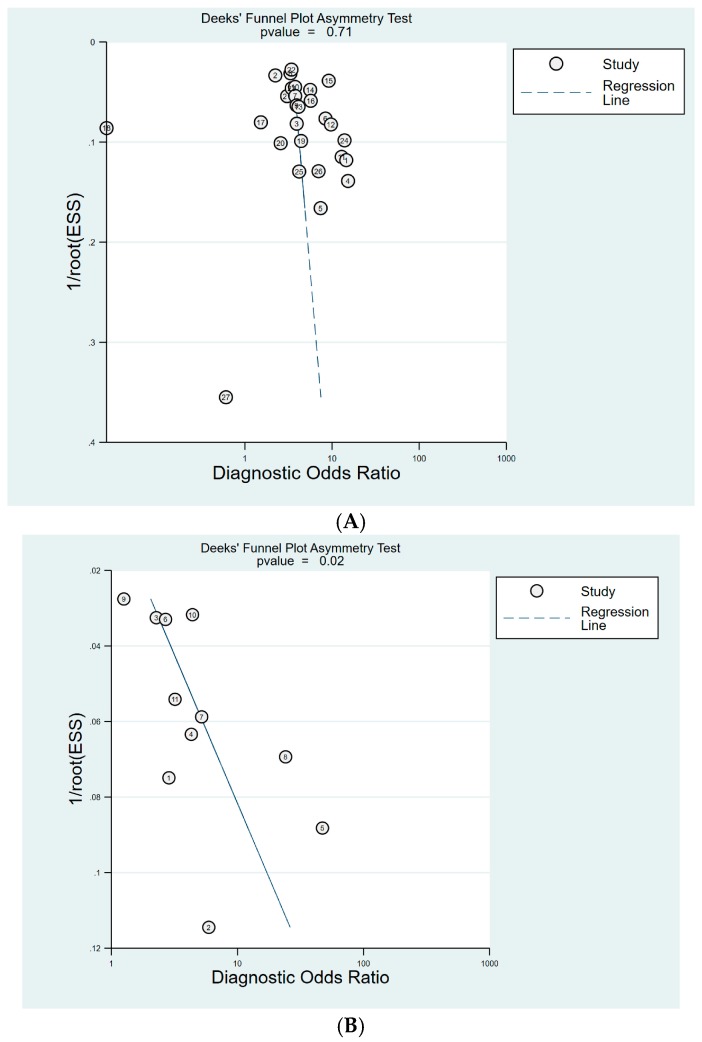
Deek’s funnel plot for the diagnosis of GC. (**A**) cut-off value with PG I ≤70 ng/mL and PG I/PG II ratio ≤3, (**B**) cut-off value with PG I/PG II ratio ≤3. GC, gastric cancer; PG, pepsinogen.

**Table 1 jcm-08-00657-t001:** Comparison of previous meta-analyses with current study.

Parameters	Current Study	Dinis-Ribeiro et al. (2004) [8]	Miki et al. (2006) [14]	Terasawa et al. (2014) [11]	Huang et al. (2015) [9]	Syrjänen et al. (2016) [13]	Zagari et al. (2017) [12]	Liu et al. (2019) [10]
Number of included studies	9 studies for the diagnosis of CAG and 17 studies for the diagnosis of GC	25 studies or book chapters for the diagnosis of GC	42 studies for the diagnosis of GC	12 studies for the diagnosis of GC	16 studies for the diagnosis of CAG and 15 studies for the diagnosis of GC	27 studies for the diagnosis of CAG	20 studies for the diagnosis of CAG	19 studies for the diagnosis of GC
Main outcome	Diagnostic validity of sPGA	Diagnostic validity of sPGA	Diagnostic validity of sPGA	Diagnostic validity of sPGA with *Helicobacter pylori* seropositivity	Diagnostic validity of sPGA	Diagnostic validity of GastroPanel (pepsinogen, gastrin-17, anti *H. pylori* antibodies)	Diagnostic validity of GastroPanel	
Searching strategy	PubMed, Embase, and the Cochrane Library (only studies in English)	PubMed and data reports from Japan (there was no information about searching keywords, the date of searching, the number of authors who performed searching, or how they managed disagreement or discrepancy of searching between authors)	PubMed and data reports from Japan (there was no information about searching keywords, the date of searching, the number of authors who performed searching, or how they managed disagreement or discrepancy of searching between authors)	PubMed, Web of Science, the Cochrane Library, and Japanese Medical Research Database (only studies in English or Japanese). The search was updated through citation-tracking	PubMed, Embase, and the CNKI (only studies in English or Chinese). Several articles were omitted.	MEDLINE (no language limitation)	PubMed, Embase, Scopus, and the Cochrane Library	PubMed, Embase, the Cochrane Library, CNKI, WanFang, VIP, and CBM databases (only studies in English or Chinese). Several articles were omitted.
Cut-off value	PG I ≤70 ng/mL and/or PG I/II ≤3	PG I ≤70 ng/mL and PG I/II ratio ≤3, PG I ≤50 ng/mL and PG I/II ratio ≤3, PG I ≤30 ng/mL and PG I/II ratio ≤2	PG I ≤70 ng/mL and PG I/II ratio ≤3	PG I ≤70 ng/mL and PG I/II ratio ≤3	Diagnostic values with various cut-off standards were pooled in a single outcome			Diagnostic values with various cut-off standards were pooled in a single outcome
Inaccurate calculation (coding) of TP/FP/FN/TN		Unknown (crude value of TP/FP/FN/TN in each study is not described)	Unknown (crude value of TP/FP/FN/TN in each study is not described). Many studies with different cut-off values were coded as those of PG I ≤70 ng/mL and PG I/II ratio ≤3 (intrinsic cutoff effect was assumed)	Not a meta-analysis with DTA. Hazard ratio was the effect size and conventional meta-analysis was done.	Detected in several studies	Unknown. Not a meta-analysis with diagnostic test accuracy (DTA). Sensitivity and specificity of each study was pooled using conventional meta-analysis method.		Detected in several studies
Determination of heterogeneity	Correlation coefficient between the logarithm of the sensitivity and specificity, beta of HSROC model, visual examination of the SROC curve	Chi-squared test (Cochrane Q statistic) with subgroup analysis according to the study population; (population-based study vs. GC screening in selected groups)	Chi-squared test (Cochrane Q statistic) (whether meta-regression was done or not is unknown)	*I^2^* statistics	*I^2^* statistics, correlation coefficient between sensitivity. Whether the correlation coefficient is between sensitivity and false positive rate or between sensitivity and specificity is not clearly defined.	*I^2^* statistics	Visual examination of the forest plot and SROC curve	*I^2^* statistics, Spearman correlation coefficient between the logarithm of the sensitivity and the logarithm of the (1—specificity), visual examination of the forest plot and SROC curve
Quality assessment	QUADAS-2	None	None	QUIPS-2, PROBAST	QUADAS-2	None	QUADAS-2	QUADAS
Inaccurate coding for subgroup analysis					Study design was inaccurately coded in several studies.			

CAG, chronic atrophic gastritis; GC, gastric cancer; sPGA, serum pepsinogen assay; CNKI, China National Knowledge Infrastructure; VIP, Chongqing VIP Chinese Science and Technology Periodical Database; CBM, Chinese BioMedical Database; PG, pepsinogen; TP, true positive; FP, false positive; FN, false negative; TN, true negative; HSROC, hierarchical summary receiver operating characteristic; SROC, summary receiver operating characteristic; QUADAS, Quality Assessment of Diagnostic Accuracy Studies; QUIPS, Quality In Prognosis Studies; PROBAST, Prediction model Risk of Bias Assessment tool.

**Table 2 jcm-08-00657-t002:** Searching strategy to find the relevant articles.

**<For CAG>**
**Database: MEDLINE (through PubMed)**
#1 “gastric atrophy”[tiab] OR “atrophic gastritis”[Mesh]
#2 “precancerous lesion”[tiab] OR “precancerous conditions”[Mesh]
#3 #1 OR #2
#4 “pepsinogen I”[tiab] OR “pepsinogen II”[tiab] OR “pepsinogen I/II”[tiab] OR “pepsinogens”[Mesh] OR “pepsinogen A”[tiab] OR “pepsinogen C”[tiab]
#5 #3 AND #4
#6 #5 English[Lang]
**Database: Embase**
#1 ‘gastric atrophy’:ab,ti,kw OR ‘atrophic gastritis’:ab,ti,kw OR ‘atrophic gastritis’/exp
#2 ‘precancerous lesion’:ab,ti,kw OR ‘precancerous condition’:ab,ti,kw
#3 #1 OR #2
#4 ‘pepsinogen’:ab,ti,kw OR ‘pepsinogen I’/exp OR ‘pepsinogen II’/exp OR ‘pepsinogen I/II’:ab,ti,kw OR ‘pepsinogen A’/exp OR ‘pepsinogen C’/exp
#5 #3 AND #4
#6 #5 AND ([article]/lim OR [article in press]/lim OR [review]/lim) AND [english]/lim
**Database: Cochrane Library**
#1 gastric atrophy:ab,ti,kw
#2 MeSH descriptor: [atrophic gastritis] explode all trees
#3 precancerous lesion:ab,ti,kw
#4 MeSH descriptor: [precancerous conditions] explode all trees
#5 #1 or #2 or #3 or #4
#6 pepsinogen I:ab,ti,kw or pepsinogen II:ab,ti,kw or pepsinogen A:ab,ti,kw or pepsinogen C:ab,ti,kw
#7 MeSH descriptor: [pepsinogens] explode all trees
#8 #6 or #7
#9 #5 and #8
**<For gastric neoplasm>**
**Database: MEDLINE (through PubMed)**
#1 “gastric cancer”[tiab] OR “gastric neoplasm”[tiab] OR “stomach cancer”[tiab] OR “stomach neoplasm”[tiab] OR “dysplasia”[tiab] OR “stomach neoplasms”[Mesh]
#2 “pepsinogen I”[tiab] OR “pepsinogen II”[tiab] OR “pepsinogen I/II”[tiab] OR “pepsinogens”[Mesh] OR “pepsinogen A”[tiab] OR “pepsinogen C”[tiab]
#3 #1 AND #2
#4 #3 English[Lang]
**Database: Embase**
#1 ‘gastric cancer’:ab,ti,kw OR ‘gastric neoplasm’:ab,ti,kw OR ‘dysplasia’:ab,ti,kw OR ‘stomach cancer’/exp OR ‘stomach tumor’/exp
#2 ‘’pepsinogen’:ab,ti,kw OR ‘pepsinogen I’/exp OR ‘pepsinogen II’/exp OR ‘pepsinogen I/II’:ab,ti,kw OR ‘pepsinogen A’/exp OR ‘pepsinogen C’/exp
#3 #1 AND #2
#4 #3 AND ([article]/lim OR [article in press]/lim OR [review]/lim) AND [english]/lim
**Database: Cochrane Library**
#1 gastric cancer:ab,ti,kw or gastric neoplasm:ab,ti,kw or stomach cancer:ab,ti,kw or stomach neoplasm:ab,ti,kw or dysplasia:ab,ti,kw
#2 MeSH descriptor: [stomach neoplasms] explode all trees
#3 #1 or #2
#4 pepsinogen I:ab,ti,kw or pepsinogen II:ab,ti,kw or pepsinogen A:ab,ti,kw or pepsinogen C:ab,ti,kw
#5 MeSH descriptor: [pepsinogens] explode all trees
#6 #4 or #5
#7 #3 and #6

CAG, chronic atrophic gastritis.

**Table 3 jcm-08-00657-t003:** Clinical characteristics of the included studies for the diagnosis of CAG.

Study	Study Format/Nationality	Diagnosis	Number of Patients	Number of Control	Cut-off Value	Detection Method of sPGA	Age (Years, Mean ± SD)	Gender (M/F)	Smoking	*H. pylori*	TP	FP	FN	TN
Inoue et al. (1998) [20]	Case–control/Japan	CAG (endoscopic diagnosis without histology)	117	83	PG I ≤70 ng/mL and PG I/PG II ratio ≤3	RIA	Mean 60.5 (range: 34–81)	91/109			96	21	21	62
Dinis-Ribeiro et al. (2004) [21]	Cross-sectional /Portugal	CAG with extensive IM (histopathologic evaluation of all three specimens collected demonstrated IM)	61	74	PG I/PG II ratio ≤3	ELISA	Median 61 (range: 26–75)	Male: 36.8%			40	16	21	58
Nardone et al. (2005) [22]	Case–control/Italy	CAG (updated Sydney classification)	30	64	PG I/PG II ratio ≤3	ELISA	Mean 56 (range: 38–75)	36/58		44/94 (46.8%)	9	0	21	64
Con et al. (2007) [23]	Case–control/Costa Rica	CAG (updated Sydney classification)	58	165	PG I ≤70 ng/mL and PG I/PG II ≤3	ELISA	51.17 ± 12.8	94/129		91.4% in patient with CAG, 68.5% in patient without CAG	45	64	13	101
Iijima et al. (2009) [24]	Case–control/Japan	CAG	20	142	PG I ≤70 ng/mL and PG I/PG II ratio ≤3	ELISA	Mean 55 (range: 22–79)	95/67			15	44	5	98
Leja et al. (2009) [25]	Case–control/Latvia, Lithuania, Taiwan	CAG (corpus, grade II-III in updated Sydney classification)	24	217	PG I/PG II ratio <3	ELISA	Mean 66.3 (range: 55–84)	68/173		165/241 (68.5%)	20	28	4	189
Agkoc et al. (2010) [26]	Case–control/Turkey	CAG	30	110	PG I/PG II ratio <3	RIA	CAG: 60.56 ± 11.29 (range: 36–76)	78/62			26	7	4	103
Yakut et al. (2013) [27]	Case–control/Turkey	CAG (updated Sydney classification)	45	117	PG I ≤70 ng/mL and PG I/PG II ratio ≤3	ELISA	55.07 ± 11.91	75/87	24 (14.8%)	98/162 (60.5%)	28	23	17	94
					PG I ≤70 ng/mL						28	22	17	95
					PG I/PG II ratio ≤3						26	12	19	105
Lee et al. (2014) [28]	Case–control/Korea	CAG (endoscopic diagnosis without histology)	1216	1204	PG I/PG II ratio ≤3	L-TIA	Mean 57.6	1506/1052		1541 (60.2%)	775	471	441	733
Kim et al. (2015) [29]	Cohort/Korea	CAG (updated Sydney classification) (antrum)	22/95		PG I ≤70 ng/mL and PG I/PG II ratio ≤3	L-TIA	57.7 ± 12.1	42/53		12/31 (38.7%) (in CAG) 17/64 (26.6%) (in no CAG)	5	9	17	64
		CAG (corpus)	19/95								8	6	11	70
Myint et al. (2015) [30]	Case–control /Myanmar	CAG (grade I-III in updated Sydney classification)	143	109	PG I ≤70 ng/mL and PG I/PG II ratio ≤3	ELISA	43.6 ± 14.2 (range: 13–85)	97/155		121/252	11	1	132	108
					PG I/PG II ratio ≤3						12	1	131	108
Kotachi et al. (2017) [31]	Case–control/Japan	CAG (endoscopic diagnosis without histology) (corpus)	370	170	PG I ≤70 ng/mL and PG I/PG II ratio ≤3	ELISA	Mean 61.2	375/165		217/540	163	0	207	170
Leja et al. (2017) [32]	Case–control/Latvia	CAG (grade II-III in updated Sydney classification)	50	755	PG I ≤70 ng/mL and PG I/PG II ratio ≤3	L-TIA	Median 51 (range: 18–88)	29% male			38	235	12	520
Loong et al. (2017) [33]	Cross-sectional /Malaysia	CAG or IM (updated Sydney classification) (corpus)	37	35	PG I ≤70 ng/mL	ELISA	56.2 ± 16.2	33/39			3	6	3	60
					PG I/PG II ratio ≤3						1	2	5	64

sPGA, serum pepsinogen assay; M, male; F, female; SD, standard deviation; TP, true positive; FP, false positive; FN, false negative; TN, true negative; CAG, chronic atrophic gastritis; PG, pepsinogen; RIA, radioimmunoassay; IM, intestinal metaplasia; ELISA, enzyme-linked immunosorbent assay; L-TIA, latex-enhanced turbidimetric immunoassay.

**Table 4 jcm-08-00657-t004:** Clinical characteristics of the included studies for the diagnosis of GC.

Study	Study Format/Nationality	Diagnosis	Number of Patients	Number of Control	Cut-off Value	Detection Method of sPGA	Age (Years, Mean ± SD)	Gender (M/F)	Smoking	*H. pylori*	TP	FP	FN	TN
Chang et al. (1992) [34]	Case–control/Taiwan	GC	192 (175 AGC)	70	PG I ≤70 ng/mL	RIA	GC: 64.6 ± 8.3/control: 51.2 ± 11.2, range: 32–85	235/27	112/262 (42.7%)		124	12	68	58
Hattori et al. (1995) [35]	Cohort/Japan (follow-up duration: 1 year)	GC (100% adenocarcinoma; 7 undifferentiated- and 11 differentiated-type histology) (sPGA positive subjects were screened by endoscopy)	18/4876		PG I ≤70 ng/mL and PG I/PG II ratio ≤3	RIA	Range: 40–61	4761/115			15	1243	3	3615
Kodoi et al. (1995) [36]	Case–control/Japan	GC	269 (127 EGCs, 142 AGCs/167 differentiated-, 102 undifferentiated-type histology)	1345 (sex, age matched)	PG I<70 ng/mL and PG I/PG II ratio <3	RIA	GC: median 65 (range: 24–80)	1080/534			162	543	107	802
Watanabe et al. (1997) [37]	Nested case–control/Japan	GC	45	225 (sex-, age-, and address-matched control)	PG I ≤70 ng/mL and PG I/PG II ratio ≤3	RIA		156/114		211/270	34	99	11	126
Kitahara et al. (1999) [38]	Cross-sectional /Japan	GC	13/5113		PG I ≤70 ng/mL and PG I/PG II ratio ≤3	RIA	Mean 52.5	2456/2657			11	1352	2	3748
					PG I/PG II ratio ≤3						11	1673	2	3427
Dinis-Ribeiro et al. (2004) [21]	Cross-sectional /Portugal	LGD	23/136		PG I/PG II ratio ≤3	ELISA	Median 61 (range: 26–75)	50/86			16	39	7	74
Ohata et al. (2004) [39]	Cohort/Japan (follow-up duration: mean 7.7 ± 0.9 year)	GC (Those with positive double-contrast barium X-ray and/or a positive PG test were further examined by endoscopy)	45/4655		PG I/PG II ratio <3	RIA	49.5 ± 4.6	100% male		3657/4655 (78.6%)	27	1585	18	3025
Kim et al. (2005) [40]	Case–control/Korea	GC	13	30	PG I<70 ng/mL and PG I/PG II ratio <3	RIA	Normal endoscopy group: mean 33.4, atrophic gastritis: 47.8, GC: 57				9	7	4	23
Watabe et al. (2005) [41]	Cohort/Japan (follow-up duration: mean 4.7 ± 1.7 years)	GC	43 (34 intestinal- and 9 diffuse-type histology)/6983		PG I ≤70 ng/mL and PG I/PG II ratio ≤3	RIA	48.9 ± 8.5	4782/2201		3216/6983 (46.1%) in total, 29/43 (67.4%) in GC	30	1495	13	5445
Oishi et al. (2006) [42]	Cohort/Japan (follow-up duration: 14 years)	GC	89/2446		PG I ≤70 ng/mL and PG I/PG II ratio ≤3	RIA	Mean 57 in male and 59 in female	1016/1430	80.2% in male and 8.2% in female	1745/2446 (71.3%) in total and 78/89 (87.6%) in GC	53	661	36	1696
Sasazuki et al. (2006) [43]	Nested case-control/Japan	GC (299 differentiated- and 159 undifferentiated-type histology)	511	511 (matched for gender, age, study area, blood donation date, fasting time at blood donation)	PG I ≤70 ng/mL and PG I/PG II ratio ≤3	EIA	57.4 ± 0.32	Male: 66.8%	GC: 35.7%, control: 30.3%		419	295	92	216
Sugiu et al. (2006) [44]	Case–control/Japan	GC	27	65	PG I/PG II ratio ≤3	RIA	Mean 57.9 (range: 15–88)	54/38		100%	23	32	4	33
Kang et al. (2008) [45]	Case–control/Korea	GC	380	626	PG I ≤70 ng/m	L-TIA	57.6 ± 13.2	585/421		788/1006 (78.3%)	275	500	105	126
		GC	380	626	PG I/PG II ratio ≤3						225	244	155	382
		Dysplasia	107	899	PG I ≤70 ng/m						88	717	19	182
		Dysplasia	107	899	PG I/PG II ratio ≤3						66	351	41	548
Yanaoka et al. (2008) [46]	Cohort/Japan (follow-up duration: mean 9.7 ± 0.9 years )	GC (Those with positive double-contrast barium X-ray and/or a positive PG test were further examined by endoscopy)	63/5209		PG I ≤70 ng/mL and PG I/PG II ratio ≤3	RIA	49.2 ± 4.7	100% male			37	1370	26	3776
Yanaoka et al. (2008) [47]	Cohort/Japan (follow-up duration: mean 9.7 ± 0.9 years )	GC (Those with positive double-contrast barium X-ray and/or a positive PG test were further examined by endoscopy)	63/5209		PG I/PG II ratio ≤3	RIA	49.2 ± 4.7	100% male		3656/5209	43	1713	20	3433
Miki et al. (2009) [48]	Cohort/Japan (follow-up duration: 15 year)	GC including intramucosal cancers (Those with a positive PG test and those with a negative PG test took endoscopy every 2 and 5 years, respectively)	125 (28 EGCs, 72 intramucosal cancers, 25 AGCs)/13789		PG I ≤70 ng/mL and PG I/PG II ratio ≤3	RIA or L-TIA	Mean 48.7	Initial enrollment: 101,892 (85,578/16,314)			110	9026	15	4638
Mizuno et al. (2009) [49]	Cohort/Japan (follow-up duration: 1 year)	GC (PG I level of ≤30 ng/mL and a PG I/PG II ratio of ≤2.0 or those with abnormal X-ray findings were advised to undergo endoscopy)	19/12120		PG I ≤70 ng/mL and PG I/PG II ratio ≤3	CLIA	Male: median 50 (range: 15–84), Female: median 49 (range: 22–84)	7590/4530			13	1743	6	10,358
Yamaji et al. (2009) [50]	Cohort/Japan (follow-up duration: mean 4.79 years)	GC	37/6158		PG I ≤70 ng/mL and PG I/PG II ratio ≤3	ELISA	Mean 49	4259/1899	2177/6158 (current or past smoker)	2901/6158	27	1333	10	4788
Yanaoka et al. (2009) [51]	Cohort/Japan (follow-up duration: mean 9.3 ± 0.7 years)	GC (Those with positive double-contrast barium X-ray and/or a positive PG test were further examined by endoscopy)	60 (40 intestinal- and 20 diffuse-type histology)/4129 (3,656 with persistent *H. pylori* infection and 473 with successful *H. pylori* eradication)		PG I ≤70 ng/mL and PG I/PG II ratio ≤3	RIA	49.8 ± 4.6 in *H. pylori* infection group, 49.6 ± 5.5 in eradication group	100% male	57.1% in *H. pylori* infection group, 55.4% in eradication group	100% infected	28	1050	32	3019
Agkoc et al. (2010) [26]	Case–control/Turkey	GC	50	90	PG I/PG II ratio <3	RIA	GC: 65.42 ± 10.28 (range: 38–83)	78/62			42	9	8	81
Kwak et al. (2010) [52]	Cross-sectional/Korea	GC	460	460	PG I/PG II ratio ≤3	L-TIA	Mean 57.9	528/392		765 (83.2%)	244	136	216	324
Mizun et al. (2010) [53]	Cohort/Japan (follow-up duration: median 9.3 years)	GC	61/2859		PG I ≤70 ng/mL and PG I/PG II ratio ≤3	RIA	55–74 category is most prevalent	1011/1848		2148/2859	44	1079	17	1719
Aikou et al. (2011) [54]	Case–control/Japan	GC	183 (107 AGCs, 76 EGCs; 86 differentiated- and 97 undifferentiated-type EGCs)	269	PG I<70 ng/mL and PG I/PG II ratio <3	ELISA	GC: 66.0 ± 10.7/control: 50.1 ± 9.9	362/90		GC: 62.3%, control: 34.9%	82	34	101	235
Chang et al. (2011) [55]	Case–control/Korea	Gastric neoplasms	297 (61 LGDs, 21 HGDs, 84 EGCs, 131 AGCs)	293	PG I/PG II ratio ≤3	L-TIA	LGD: 60.2 ± 9.5, HGD: 63.1 ± 8.6, EGC: 59.8 ± 9.2, AGC: 61.6 ± 12.6, control: 50.7 ± 13.6	368/222	Gastric neoplasms: 22.8–51.9%, control: 24.2%	Gastric neoplasms: 60.7–81%, control: 58%	184	89	113	204
Kaise et al. (2011) [56]	Case–control/Japan	GC	192	1254	PG I ≤70 ng/mL and PG I/PG II ratio ≤3	CLIA	GC: 64.3 ± 9.7/control: 52.3 ± 12.4	GC: 5:1, control: 1.2:1	GC: 63%, control: 38.2%	GC: 83.9%, control: 30.1%	129	229	63	1025
Kikuchi et al. (2011) [57]	Case–control/Japan	EGC	122 (114 well- to moderate-differentiated EGCs and 8 poorly-differentiated EGCs)	178	PG I ≤70 ng/mL and PG I/PG II ratio ≤3	CLIA	GC: 68.2 ± 9.7/control: 56.2 ± 14.9	187/113		GC: 100/122 (82%), control: 109/178 (61.2%)	95	68	27	110
					PG I ≤70 ng/mL						114	148	8	30
					PG I/PG II ratio ≤3						100	83	22	95
Ito et al. (2012) [58]	Case–control/Japan	Diffuse-type EGC	42	511	PG I ≤70 ng/mL and PG I/PG II ratio ≤3	RIA	GC: mean 57.2 in male, 59.1 in female. Control: mean 58.5	305/248		387/553	20	191	22	320
Lomba-Viana et al. (2012) [59]	Cohort/Portugal (follow-up duration: 3–5 year)	GC	6 (5 intestinal- and 1 diffuse-type histology/3 EGCs and 3 AGCs)/514		PG I ≤70 ng/mL and PG I/PG II ratio ≤3	ELISA	Median 60 (range: 40–79)	76/438		165/514 (32.1%)	6	268	3	237
Zhang et al. (2012) [60]	Cohort/China (follow-up duration: 14 years)	GC	26/1501		PG I ≤70 ng/mL and PG I/PG II ratio ≤3	RIA	45.29 ± 12.18	554/947		995/1501 (66.3%)	9	158	17	1317
Yakut et al. (2013) [27]	Case–control/Turkey	Dysplasia	37	125	PG I ≤70 ng/mL and PG I/PG II ratio ≤3	ELISA	57.52 ± 11.16	75/87	24 (14.8%)	98/162 (60.5%)	13	38	24	87
					PG I ≤70 ng/mL						13	37	24	88
					PG I/PG II ratio ≤3						8	30	29	95
Choi et al. (2014) [61]	Case–control/Korea	Gastric neoplasms	17	3311	PG I ≤70 ng/mL and PG I/PG II ratio ≤3	L-TIA	Mean 49.8–59.0	1979/1349			9	438	8	2873
Huang et al. (2014) [62]	Nested case-control/China	GC	72	37	PG I ≤70 ng/mL and PG I/PG II ratio ≤3	CLIA	GC: 61.7 ± 1.4, control: 56.7 ± 2.8	GC: 1.23:1. Control: 1.31:1		GC: 66.7%, control: 48%	27	7	45	30
Yoshida et al. (2014) [63]	Cohort/Japan (follow-up duration: mean 11.6 ± 4.3 years)	GC (those with positive double-contrast barium X-ray and/or a positive PG test were further examined by endoscopy)	87/4655		PG I ≤70 ng/mL and PG I/PG II ratio ≤3	RIA	49.5 ± 4.6	100% male	59.3%	3657/4655	48	1314	39	3254
Zhang et al. (2014) [64]	Case–control/China	GC	82 (69 AGCs, 13 EGCs)	142	PG I ≤70 ng/mL	ELISA	Patients with gastrointestinal diseases: 52.3 ± 12.3 (range 19–80), control: 52.4 ± 15.1 (range 29–77)	163/85	85.4% in patients with GC, 74.4% in control		56	25	26	117
					PG I/PG II ratio ≤3						21	2	61	140
Eybpoosh et al. (2015) [65]	Cross-sectional/Iran	GC	578 (62 EGCs, 516 AGCs/315 intestinal-, 203 diffuse-, 69 mixed-type histology/274 undifferentiated-, 304 differentiated-type histology)	763	PG I ≤70 ng/mL and PG I/PG II ratio ≤3	ELISA		750/591	399/1341		100	44	478	719
					PG I ≤70 ng/mL						133	234	445	529
					PG I/PG II ratio ≤3						32	34	546	729
Ikeda et al. (2016) [66]	Cohort/Japan (follow-up: at least 20 years)	GC	123/2446		PG I ≤70 ng/mL and PG I/PG II ratio ≤3	RIA	58.3 ± 11.4	1016/1430	24.6%	1761/2446	70	644	53	1679
Cho et al. (2017) [67]	Case–control/Korea	Gastric neoplasms	87 (19 LGDs, 16 HGDs, 40 EGCs, 12 AGCs)	311	PG I/PG II ratio ≤3	L-TIA	48.2 ± 16.6	170/228		209/398 (52.5%) (46% with neoplasm vs. 75.9% without neoplasm)	59	62	28	249
Hamashima et al. (2017) [68]	Nested case-control/Japan	GC	497	497 (matched for sex, age, blood donation date, and fasting time at blood donation)	PG I/PG II ratio ≤3	EIA	57.5 ± 7.2	Male: 66.4%			432	299	65	198
Tu et al. (2017) [69]	Cohort/China (follow-up duration: median 11.6 years)	GC	86/12018		PG I ≤70 ng/mL	ELISA	GC: 59.0 ± 10.6/GC-free: 49.6 ± 10.7	82.6% male in GC/45.1% male in GC-free	39% in GC/36.4% in GC-free		27	3642	59	8290
					PG I/PG II ratio <3						15	728	71	11204
Castro et al. (2018) [70]	Cohort/Portugal (follow-up duration: median 6.5 years for sPGA (+)/7.5 years for sPGA (–)	GC (100% adenocarcinoma)	26/5913		PG I ≤70 ng/mL and PG I/PG II ratio ≤3	ELISA	Range: 40–74	2257/3656			9	216	17	5671
Kwak et al. (2018) [71]	Cohort/Korea (follow-up duration: mean 5.6 years)	GC	15/3297		PG I ≤70 ng/mL and PG I/PG II ratio ≤3	L-TIA	51.3 ± 9.4	2326/971		2020/3297	7	567	8	2715
		Gastric neoplasms	29/3297								12	562	17	2706
Lee et al. (2018) [72]	Case–control/Korea	EGC	30	30	PG I<70 ng/mL and PG I/PG II ratio <3	L-TIA	59.5 ± 10.7 (patients with EGC) vs. 66.6 ± 12.0 (control)	36/24			10	2	20	28
Sjomina et al. (2018) [73]	Cross-sectional/Latvia	GC	2	257	PG I<70 ng/mL and PG I/PG II ratio <3	L-TIA	56.5 ± 12.5	82/177		177 (66%)	1	160	1	97
		Gastric dysplasia	21	238							17	144	4	94

sPGA, serum pepsinogen assay; M, male; F, female; SD, standard deviation; TP, true positive; FP, false positive; FN, false negative; TN, true negative; GC, gastric cancer; EGC, early gastric cancer; AGC, advanced gastric cancer; LGD, low grade dysplasia; HGD, high grade dysplasia; PG, pepsinogen; RIA, radioimmunoassay; EIA, enzyme immunoassay; CLIA, chemiluminescent immunoassay; ELISA, enzyme-linked immunosorbent assay; L-TIA, latex-enhanced turbidimetric immunoassay.

**Table 5 jcm-08-00657-t005:** Summary of DTA and subgroup analysis of the included studies for the diagnosis of CAG.

Subgroup	Number of Included Studies	Sensitivity (95% CI)	Specificity (95% CI)	PLR	NLR	DOR	AUC
**Cut-off value: PG I ≤70 ng/mL and PG I/PG II ratio ≤3**	**8**	**0.59 (0.38–0.78)**	**0.89 (0.70–0.97)**	**5.5 (2.3–13.0)**	**0.46 (0.30–0.69)**	**12 (6–25)**	**0.81 (0.77–0.84)**
Age (years, median or mean)							
<60	6	0.54 (0.29–0.78)	0.84 (0.64–0.94)	3.5 (2.1–5.8)	0.54 (0.35–0.84)	6 (4–10)	0.78 (0.74–0.81)
60≤	2	Null	Null	Null	Null	Null	Null
Methodological quality of included studies							
High-quality	5	0.68 (0.54–0.79)	0.76 (0.64–0.85)	2.79 (2.04–3.80)	0.43 (0.32–0.57)	7 (4–10)	0.78 (0.74–0.82)
Low-quality	3	Null	Null	Null	Null	Null	Null
**Cut-off value: PG I/PG II ratio ≤3**	**8**	**0.50 (0.28–0.72)**	**0.94 (0.82–0.98)**	**7.8 (3.3–18.1)**	**0.53 (0.34–0.82)**	**15 (6–37)**	**0.85 (0.81–0.88)**
Ethnicity							
Western	5	0.66 (0.45–0.81)	0.92 (0.81–0.97)	8.5 (3.7–19.4)	0.37 (0.22–0.62)	23 (9–57)	0.88 (0.85–0.91)
Asian	3	Null	Null	Null	Null	Null	Null
Age (years, median or mean)							
<60	5	0.31 (0.15–0.53)	0.97 (0.80–0.99)	8.9 (2.2–35.9)	0.71 (0.57–0.89)	12 (3–45)	0.67 (0.62–0.71)
60≤	3	Null	Null	Null	Null	Null	Null
Methodological quality of included studies							
High-quality	4	0.66 (0.40–0.85)	0.94 (0.84–0.98)	10.7 (4.8–24.1)	0.36 (0.18–0.71)	30 (11–78)	0.92 (0.90–0.94)
Low-quality	4	0.32 (0.12–0.62)	0.92 (0.63–0.99)	4.1 (1.4–12.3)	0.73 (0.56–0.97)	6 (2–15)	0.67 (0.63–0.71)
Total number of included patients							
<1000	7	0.49 (0.24–0.74)	0.95 (0.87–0.98)	9.6 (4.8–19.4)	0.54 (0.33–0.89)	18 (8–41)	0.90 (0.87–0.92)
1000≤	1	Null	Null	Null	Null	Null	Null

Subgroups with less than four studies were defined as null because quantitative analysis was not possible. DTA, diagnostic test accuracy; CAG, chronic atrophic gastritis; CI, confidence interval; PLR, positive likelihood ratio; NLR, negative likelihood ratio; DOR, diagnostic odds ratio; AUC, area under the curve; PG, pepsinogen. Bold: Summary DTA of the included studies for the diagnosis of CAG.

**Table 6 jcm-08-00657-t006:** Summary of DTA and subgroup analysis of the included studies for the diagnosis of GC.

Subgroup	Number of Included Studies	Sensitivity (95% CI)	Specificity (95% CI)	PLR	NLR	DOR	AUC
**Cut-off value: PG I ≤70 ng/mL and PG I/PG II ratio ≤3**	**27**	**0.59 (0.50–0.67)**	**0.73 (0.64–0.81)**	**2.2 (1.7–2.9)**	**0.56 (0.46–0.68)**	**4 (3–6)**	**0.70 (0.66–0.74)**
Ethnicity							
Asian	24	0.60 (0.52–0.68)	0.75 (0.68–0.80)	2.4 (2.0–2.8)	0.53 (0.46–0.61)	5 (4–6)	0.73 (0.69–0.77)
Western	3	null	null	null	null	null	null
Published year							
2010–2018	15	0.46 (0.35–0.57)	0.77 (0.63–0.87)	2.0 (1.2–3.4)	0.70 (0.56–0.89)	3 (1–6)	0.61 (0.57–0.66)
1995–2009	12	0.71 (0.64–0.78)	0.68 (0.59–0.76)	2.2 (1.8–2.8)	0.42 (0.34–0.52)	5 (4–8)	0.76 (0.72–0.79)
Total number of included patients							
<1000	9	0.50 (0.34–0.65)	0.65 (0.44–0.81)	1.4 (0.7–2.8)	0.78 (0.48–1.25)	2 (1–6)	0.58 (0.54–0.62)
1000≤	18	0.61 (0.50–0.70)	0.77 (0.68–0.83)	2.6 (2.1–3.2)	0.51 (0.43–0.61)	5 (4–7)	0.74 (0.70–0.78)
**Cut-off value: PG I ≤70 ng/mL**	**6**	**0.62 (0.38–0.82)**	**0.57 (0.32–0.79)**	**1.4 (0.9–2.3)**	**0.67 (0.40–1.11)**	**2 (1–5)**	**0.63 (0.58–0.67)**
Methodological quality of included studies							
High-quality	5	0.52 (0.33–0.70)	0.66 (0.43–0.83)	1.5 (0.8–2.9)	0.73 (0.47–1.16)	2 (1–6)	0.61 (0.57–0.65)
Low-quality	1	null	null	null	null	null	null
**Cut-off value: PG I/PG II ratio ≤3**	**11**	**0.56 (0.35–0.75)**	**0.78 (0.62–0.88)**	**2.5 (1.7–3.7)**	**0.56 (0.39–0.81)**	**4 (3–8)**	**0.74 (0.70–0.78)**
Ethnicity							
Asian	10	0.52 (0.30–0.73)	0.75 (0.58–0.87)	2.1 (1.7–2.7)	0.63 (0.47–0.86)	3 (2–5)	0.70 (0.66–0.74)
Western	1	null	null	null	null	null	null

Subgroups with less than four studies were defined as null because quantitative analysis was not possible. DTA, diagnostic test accuracy; GC, gastric cancer; CI, confidence interval; PLR, positive likelihood ratio; NLR, negative likelihood ratio; DOR, diagnostic odds ratio; AUC, area under the curve; PG, pepsinogen. Bold: Summary DTA of the included studies for the diagnosis of GC.

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
