# Peer review of "Prediction of Chronic Atrophic Gastritis and Gastric Neoplasms by Serum Pepsinogen Assay: A Systematic Review and Meta-Analysis of Diagnostic Test Accuracy"

_jcm, 2019, doi:10.3390/jcm8050657_

Round 1
Reviewer 1 Report
The title is too similar for a previous one yours (medicine, 2019 Jan; 98(4): e14240) please change it.
It is strange that study of ACG are null for Asian countries, could you discuss this point in the discussion.
lane 416, please introduce the concept that some researchers have proposed also a different pepsinogen cut-off in patients at risk for the consideration for gastric cancer development. In particular PG I are generally lesser in patients with autoimmune atrophic gastritis and this could be take in consideration before to evaluate the cut-off of PG I for gastric cancer risk in this particular subtype of patients and in relation to the level of gastrin G 17 (for ex. see clin trans gastroenteorl. 2016 Jul; 7(7): e183.)
lane 423 "indicate" could be modified with "confirm" since data is not new and yet reported in literature.
lane 430-437...overestimation may result also from a difference in the gastric cancer epidemiology from intestinal subtype to an increase in diffuse gastric cancer subtype ?? In particular true in Westwern countries and in recent decades ?? please discuss also to this point.
Author Response
Prediction of chronic atrophic gastritis and gastric neoplasms by serum pepsinogen assay; systematic review and meta-analysis of diagnostic test accuracy
Dear editor and reviewers
First of all, authors appreciate the editor’s and reviewers’ thoughtful and helpful comments. Also, we are pleased to have an opportunity to make this paper to be an even better one and to be accepted with revision, because the editor and reviewers provided additional important points that we haven’t realized before.
Here, we are submitting the revised manuscript that addresses several concerns of the editor and reviewers. We have included the changes as recommended by the editor and reviewers in the revised manuscript (colored as red). Although the originally submitted manuscript was edited by professional English editing company and English editing was not an issue in the first review process, the revised version might need another round of English editing. The authors can do English editing once more if requested.
We hope that this paper will now be considered for publication.
We thank you for your time and look forward to your reply.
Yours Sincerely
-------------------------------------------------------------------------------------------------------------------------------
Manuscript ID jcm-509347
Point-to-point responses to comments by the Reviewer #1
1. The title is too similar for a previous one yours (medicine, 2019 Jan; 98(4): e14240) please change it.
-------------------------------------------------------------------------------------------------------------------------------
The authors wish to thank the reviewer for offering thoughtful and helpful comments.
R1: We totally agree with the opinion of the reviewer. We changed title as ‘Prediction of chronic atrophic gastritis and gastric neoplasms by serum pepsinogen assay; systematic review and meta-analysis of diagnostic test accuracy’ (colored red). Thanks again for offering helpful comments.
2. It is strange that study of ACG are null for Asian countries, could you discuss this point in the discussion.
-------------------------------------------------------------------------------------------------------------------------------
The authors wish to thank the reviewer for offering helpful comments.
R2: We are sorry to make you confused. The reason of Asian studies are null in the Table 5 is that minimum number of studies required for the quantitative analysis is four as described in line 306 and 381-382. The number of included studies in Asian ethnicity subgroup was three. Therefore, these studies were set as null in Table 5.
To enhance the readability and to avoid the misleading, we added the following sentence in the footnote of table 5 and 6; “Subgroups with less than four studies were defined as null because quantitative analysis was not possible”. (colored red).
Thanks again for offering helpful comments.
3. lane 416, please introduce the concept that some researchers have proposed also a different pepsinogen cut-off in patients at risk for the consideration for gastric cancer development. In particular PG I are generally lesser in patients with autoimmune atrophic gastritis and this could be take in consideration before to evaluate the cut-off of PG I for gastric cancer risk in this particular subtype of patients and in relation to the level of gastrin G 17 (for ex. see clin trans gastroenteorl. 2016 Jul; 7(7): e183.)
-------------------------------------------------------------------------------------------------------------------------------
The authors wish to thank the reviewer for offering thoughtful and helpful comments.
R3: Your comment was very helpful to us. The following paragraph and reference were added in the discussion section (line 462-476); “The distribution of CAG or IM (known as pre-malignant or high-risk lesions of GC) in entire population affects the determination of optimal cut-off value of sPGA (spectrum bias). In our meta-analysis, study by Dinis-Ribeiro et al. [21] included high-risk patients of GC, such as AG, IM or dysplasia, excluding healthy population and showed higher sensitivity compared to that of pooled analysis with cut-off of PGI/PGII≤3 (0.66 vs. 0.50) (Table 5). Previous study by Valli De Re et al. [78] also included high-risk patients, such as first degree relatives of patients with GC or CAG and showed high sensitivity and specificity of 0.96 and 0.93 for the prediction of Operative Link on Gastric Intestinal Metaplasia Assessment (OLGIM) stage ≥2 with cut-off of PG I ≤47.9 ng/ml. The proposed cut-off of PG I was lower than 70 ng/ml because they included high-risk population. However, they proposed algorithm approach of using gastrin-17 first, because they included high-risk patients and gastrin-17 showed highest discrimination capacity of CAG among proposed biomarkers. For the next-step, they recommended using PG I ≤47.9 ng/ml for the prediction of OLGIM stage ≥2. PG I generally shows a low level in CAG; however, if an optimal cut-off should be determined in a high-risk population, lower cut-off value might be required. And a combination with a marker, such as gastrin-17, which shows high discriminative performance of CAG could be considered.”
Thanks again for offering helpful comments.
4. lane 423 "indicate" could be modified with "confirm" since data is not new and yet reported in literature.
-------------------------------------------------------------------------------------------------------------------------------
The authors wish to thank the reviewer for offering helpful comments.
R4: The terminology ‘indicate’ was corrected to ‘confirm’ as reviewer’s suggestion (line 429). We appreciate the reviewer’s thoughtful comments.
5. lane 430-437...overestimation may result also from a difference in the gastric cancer epidemiology from intestinal subtype to an increase in diffuse gastric cancer subtype ?? In particular true in Westwern countries and in recent decades ?? please discuss also to this point.
-------------------------------------------------------------------------------------------------------------------------------
The authors wish to thank the reviewer for offering helpful comments.
R5: We are sorry to make a confusion. In table 5, although studies with Western population showed slightly higher AUC (0.88 vs. 0.85) than pooled AUC, the value is closer to that of high-quality studies subgroup (0.92), indicating it is not an overestimation, rather we need more Western population data to enhance the level of evidence. In table 6, recently published subgroup showed much lower AUC (0.61 vs. 0.76) than that of old publications; however, the AUC of recently published subgroup was closer to that of high-quality subgroup (0.68; data not shown because it was not a source of heterogeneity in meta-regression), indicating overestimation of old publications. To enhance the readability and to avoid misleading, we added sentences stated above in the discussion section (line 452-458) (colored red).
In terms of the diffuse-type GC, we could not evaluate the effect of histologic type because only several studies presented this information as shown in Table 4. PG I and PG I/II ratio were found to be more valuable for intestinal-type than diffuse-type GC, which is probably related with different grades of CAG and IM according to histologic type. If the ratio of diffuse-type GC is higher then the efficacy of sPGA could be lower. This could be also one of the limitation of this study and content was added in the discussion section (linie 481-482) (colored red).
Reviewer 2 Report
This study shows that the performance of sPGA is better for the diagnosis of chronic atrophic gastritis than gastric cancer as well as the diagnostic validity of sPGA with cut-off value of PG I/PG II ≤3.
In the sentence from line 45 to line 49 ".....serum 47 pepsinogen assay (sPGA), which reveals concentration of pepsinogen I (PG I) and ratio of PG I/PG II, 48 has been accepted as a non-invasive test for predicting CAG or GC..."the term accepted should be replaced with the term "proposed". The same in the sentence from line 50 to line 51.
Could the authors specify "exactly", in the text, the weywords used in this research?
Line 251 "Fiver" should be "five".
Author Response
Prediction of chronic atrophic gastritis and gastric neoplasms by serum pepsinogen assay; systematic review and meta-analysis of diagnostic test accuracy
Dear editor and reviewers
First of all, authors appreciate the editor’s and reviewers’ thoughtful and helpful comments. Also, we are pleased to have an opportunity to make this paper to be an even better one and to be accepted with revision, because the editor and reviewers provided additional important points that we haven’t realized before.
Here, we are submitting the revised manuscript that addresses several concerns of the editor and reviewers. We have included the changes as recommended by the editor and reviewers in the revised manuscript (colored as red). Although the originally submitted manuscript was edited by professional English editing company and English editing was not an issue in the first review process, the revised version might need another round of English editing. The authors can do English editing once more if requested.
We hope that this paper will now be considered for publication.
We thank you for your time and look forward to your reply.
Yours Sincerely
---------------------------------------------
Manuscript ID jcm-509347
Point-to-point responses to comments by the Reviewer #2
1. In the sentence from line 45 to line 49 ".....serum 47 pepsinogen assay (sPGA), which reveals concentration of pepsinogen I (PG I) and ratio of PG I/PG II, 48 has been accepted as a non-invasive test for predicting CAG or GC..."the term accepted should be replaced with the term "proposed". The same in the sentence from line 50 to line 51.
-------------------------------------------------------------------------------------------------------------------------------
The authors wish to thank the reviewer for offering thoughtful and helpful comments.
R1: Authors agree with the reviewer’s point. We changed terminology from “accepted” to “proposed” in the whole manuscript (line 24, 52, 55, and 414) (colored red). Thanks again for offering helpful comments.
2. Could the authors specify "exactly", in the text, the weywords used in this research?
-------------------------------------------------------------------------------------------------------------------------------
The authors wish to thank the reviewer for offering helpful comments.
R2: We are sorry to make you confused. There are three ‘exactly’ terminology in Table 1. All the “not exactly defined or described” were meant to be “not clearly defined” in Table 1.
Study by Dinis-Ribeiro M et al. [8] described searching strategy in the manuscript. However, the process cannot be reproduced because there was no information about searching keywords, the date of searching, the number of authors who performed searching, or how they managed disagreement or discrepancy of searching between authors, etc.. Study by Miki et al. [14] also described same searching strategy with that of study by Dinis-Ribeiro M et al.
There are several methods of analysis for the determination of heterogeneity. Spearman correlation analysis between sensitivity and false positive rate or between sensitivity and specificity is the representative method in the diagnostic test accuracy meta-analysis. Although, study by Huang YK et al. [9] described that they used spearman correlation analysis, there was no clear explanation about whether correlation coefficient is between sensitivity and false positive rate or between sensitivity and specificity.
Therefore, the following sentences were added in the Table 1 (colored red); “there was no information about searching keywords, the date of searching, the number of authors who performed searching, or how they managed disagreement or discrepancy of searching between authors”, “whether correlation coefficient is between sensitivity and false positive rate or between sensitivity and specificity is not clearly defined”.
Thanks again for offering helpful comments.
3. Line 251 "Fiver" should be "five".
-------------------------------------------------------------------------------------------------------------------------------
The authors wish to thank the reviewer for offering thoughtful and helpful comments.
R3: The mistyping was corrected (colored red) and the entire text was reviewed again. Thanks again for offering helpful comment.